# On Sample Optimality in Personalized Collaborative and Federated Learning

**Mathieu Even[1], Laurent Massoulié [1,2], Kevin Scaman [1]**

[1]Inria Paris - Département d'informatique de l'ENS, PSL Research University

[2]Microsoft-Inria Joint Center

## Abstract

In personalized federated learning, each member of a potentially large set of agents aims to train a model minimizing its loss function averaged over its local data distribution. We study this problem under the lens of stochastic optimization, focusing on a scenario with a large number of agents, that each possess very few data samples from their local data distribution. Specifically, we prove novel matching lower and upper bounds on the number of samples required from all agents to approximately minimize the generalization error of a fixed agent. We provide strategies matching these lower bounds, based on a *gradient filtering* approach: given prior knowledge on some notion of distance between local data distributions, agents filter and aggregate stochastic gradients received from other agents, in order to achieve an optimal bias-variance trade-off. Finally, we quantify the impact of using rough estimations of the distances between local distributions of agents, based on a very small number of local samples.

## 1   Introduction

A central task in federated learning [31, 40] is the training of a common model from local data sets held by individual agents. A typical application is when users (*e.g.* mobile phones, hospitals) want to make predictions (*e.g.* next-word prediction, treatment prescriptions), but each has access to very few data samples, hence the need for collaboration. As highlighted by many recent works (*e.g.* Hanzely et al. [28], Mansour et al. [39]), while training a global model yields better statistical efficiency on the combined datasets of all agents by increasing the number of samples linearly in the number of agents, this approach can suffer from a dramatically poor generalization error on local datasets. A solution to this generalization issue is the training of *personalized* models, a midway between a shared model between agents and models trained locally without any coordination.

An ideal approach would take the best of both worlds: increased statistical efficiency by using more samples, while keeping local generalization errors low. This raises the fundamental question: what is the optimal bias/variance tradeoff between personalization and coordination, and how can it be achieved?

We formulate the personalized federated learning problem as follows, studying it under the lens of stochastic optimization [5]. Consider $N \in \mathbb{N}^*$ agents denoted by integers $1 \leqslant i \leqslant N$, each desiring to minimize its own local function $f_i : \mathbb{R}^d \to \mathbb{R}$, while sharing their stochastic gradients. Since only a limited number of samples are locally available, we focus on *stochastic gradient descent*-like algorithms, where agents each sequentially compute stochastic gradients $g_i^k$ such that $\mathbb{E}\left[g_i^k\right] = \nabla f_i$. In order to reduce the sample complexity, *i.e.* the number of samples or stochastic gradients required to reach small generalization error, agents thus need to use stochastic gradients from other agents, that are *biased* since in general $\mathbb{E}\left[g_i^k\right] \neq \nabla f_j$. Our algorithms are based on a *gradient filtering* approach:

36th Conference on Neural Information Processing Systems (NeurIPS 2022).

upon reception of stochastic gradients $(g_j^k)_j$, agent $i$ *filters* these gradients and aggregates them using some weights $\lambda_j$ into $\sum_j \lambda_j g_j^k$, in order to achieve some bias/variance trade-off.

## 1.1 Contributions and outline of the paper

In this paper, we consider an oracle model where at each step $k = 1, 2, \ldots$, all agents may draw a sample according to their local distribution. We aim at computing the number of stochastic gradients sampled from all agents, required to reach a small generalization error, in terms of biases (distances between functions or distributions), regularity, and noise assumptions. The oracle model, main assumptions and problem formulations are given in Section 2. Our main contributions are then as follows.

*(i)* In Section 3 we prove *information theoretic* lower bounds: to reach a target generalization error $\varepsilon > 0$ for a fixed agent $i$, no algorithm can achieve a reduction in the number of oracle calls by a factor larger than the total number of agents $\varepsilon$-close –in a suitable sense– to agent $i$.

*(ii)* We next study a naive strategy based on weighted gradient averaging algorithms, coined *all-for-one*, that matches this lower bound, at the cost of high communication and storage requirements.

*(iii)* We then propose in Section 5 a parallel extension of the simple weighted gradient averaging algorithm that yields an efficient algorithm for collaborative generalization error minimization problems. In this algorithm, agents compute stochastic gradients at *their* local estimate, and broadcast it to other agents who may use these to update their own estimates. For $x^k = (x_1^k, \ldots, x_N^k)$ where $x_i^k$ is the local estimate of agent $i$ at iteration $k$, updates of the ALL-FOR-ALL algorithm write as:

$$x^{k+1} = x^k - \eta W g^k \,,$$

where $g^k = (g_1^k, ..., g_N^k)$ for an unbiased stochastic gradient $g_i^k$ of function $f_i$, a step size $\eta$, and a carefully chosen symmetric matrix $W$. Agent $i$ thus uses stochastic gradients that are doubly biased, as gradients of a "wrong function" $f_j$ instead of $f_i$ computed at a "wrong location" $x_j^k$ instead of $x_i^k$. Interestingly, note that the ALL-FOR-ALL algorithm is not a gossip algorithm *per se* (see *e.g.* [45]), since the matrix $W$ is not doubly-stochastic: gradients are not aggregated with weights that sum to $1$. Moreover, $W$ depends on the distance between local agents distributions, and thus requires either prior information on the local distributions, or estimating these distances as a pre-processing step.

*(iv)* We finally study in Section 6 the impact of estimating, based on a very limited number of samples, the matrix $W$ to use in the ALL-FOR-ALL algorithm. Under a mixture model assumption on the agents, we obtain that for a bounded – up to logarithmic factors – number of samples per agent, any arbitrary small generalization can be reached, with an optimal collaboration speedup in terms of the number of agents in each mixture of the mixture model.

## 1.2 Related works

*Federated Learning* is a paradigm in machine learning where training is done collaboratively among several agents, taking into account privacy constraints [31, 36, 40, 51]. A central task is the training of a common model for all agents, for which both *centralized* approaches orchestrated by a server and *decentralized* approaches with no central coordinator [44] have been considered. The algorithms we propose in this paper are well suited for a decentralized implementation.

As observed in Hanzely et al. [28], training a common model for all users can lead to poor generalization on certain tasks such as e.g. next-word prediction. To improve both accuracy and fairness, *personalized* models thus need to be learnt for each agent [38, 43, 54]. Approaches to personalization include fine-tuning [10, 34], transfer learning techniques [19, 49, 51], using shared-representation models [11]. Personalization in FL can also be formulated as the training of local models with a regularization term that enforces collaboration between users [28] or with a meta-learning approach [9, 25, 30]. We refer the interested reader to Kulkarni et al. [37] for a broader survey of Personalized Federated Learning.

While the goal of personalization is to minimize local generalization errors, the above cited works do not provide theoretical guarantees over the sample complexity to obtain small local errors, but instead control errors on a regularized problem, in terms of communication rounds or full gradients used, and not in terms of samples used. Deng et al. [14], Mansour et al. [39] among others provide generalization errors under a statistical learning framework that depend on VC-dimensions and

on distances between each local data distribution and the mixture of all datasets. Donahue and Kleinberg [16, 17] study the bias-variance trade-off between collaboration and personalization for mean estimation in a game-theoretic framework. Beaussart et al. [3], Chayti et al. [8], Grimberg et al. [27] also concurrently frame personalization as a stochastic optimization problem with biased gradients and are the works closest to ours. They consider the training of a single agent with biased gradients from another group of agents dedicated to this agent, and obtain performance guarantees in terms of distance between individual function $f_i$ and the average $N^{-1} \sum_j f_j$. In contrast, we obtain more general performance bounds based on distance bounds between all pairs of functions $f_i, f_j$ (or equivalently, pairs of local distributions), in the case where all agents desire to minimize their local objective; our "bias assumption" is also milder. In addition, we prove matching lower bounds, and study under a mixture model the statistical efficiency of our approach.

Concurrently, [15] use similar dissimilarity notions (Integral Probability Metrics, albeit with respect to different function spaces) to show upper-bounds that closely resemble ours of Theorem 3, when training over mixtures of distributions (without algorithmic solutions as the *all-for-all* algorithm), together with insights on who to collaborate with for agents. These results, obtained in a different framework than ours (hypothesis and VC-dimension bounds, rather than our stochastic optimization framework), fall into the frame of our lower bound (and match it up to constant factors), form an orthogonal line of work.

Finally, data-heterogeneity has long been a challenge in Federated optimization, as for instance noted in the analyses and performances of the Local SGD algorithm [33, 53]. Many algorithmic solutions have been proposed to counterweight this effect [32, 42] (non-exhaustive list). Yet this line of work studies the effect of data-heterogeneity on the convergence guarantees of FL algorithms that train *one* global model, irrespectively of the local generalization property of this trained global model. Our work is orthogonal, and focuses on data-heterogeneity as a challenge for statistical meaning (local generalization) of the model(s) trained, as opposed to related works that study data-heterogeneity as a challenge in distributed or federated learning to design fast and scalable algorithms. Putting into perspective these two views on the challenge data-heterogeneity in FL seems however necessary, and stresses its importance.

## 2 Problem Statement and Assumptions

We now detail our objectives and the necessary technical assumptions. We consider general stochastic gradient methods and formulate our problem, assumptions and algorithms accordingly.

### 2.1 Problem setting

Let $\mathcal{D}_i$ for $1 \leqslant i \leqslant N$ be a probability distribution on a set $\Xi$ (agent $i$'s local distribution, *not* its empirical distribution), $\ell : \mathbb{R}^d \times \Xi \to \mathbb{R}$ a loss function. We assume that the function $f_i$ that agent $i$ aims at minimizing is the generalization error on agent $i$'s local distribution:

$$f_i(x) = \mathbb{E}_{\xi_i \sim \mathcal{D}_i} \left[ \ell(x, \xi_i) \right], \quad x \in \mathbb{R}^d . \tag{1}$$

We coin this problem as *collaborative generalization error minimization (GEM)*. At every iteration $k = 1, 2, \ldots$, agent $i$ may access unbiased *i.i.d.* estimates $g_i^k(x)$ of $\nabla f_i(x)$:

$$g_i^k(x) = \nabla_x \ell(x, \xi_i^k), \quad \xi_i^k \sim \mathcal{D}_i, \quad x \in \mathbb{R}^d, 1 \leqslant i \leqslant N .$$

Counting the number of stochastic gradients used in the whole set of agents to reach a precision $\varepsilon$ for $f_i$ thus reduces to computing the number of samples required from all agents to obtain local generalization error $\varepsilon$ for agent $i$. To specify the information shared between agents via access to stochastic gradients, we define the following oracle, that lets at every iteration all agents sample a stochastic gradient. After $K$ oracle queries, each agent will have sampled $K$ stochastic gradients for a total of $NK$ in the whole set of agents. Let $\left\{ (\xi_1^k, \ldots, \xi_N^k), k \geqslant 0 \right\}$ a sequence of *i.i.d.* random variables of law $\mathcal{D}_1 \times \cdots \times \mathcal{D}_N$. Given the initial shared knowledge $\mathcal{S}_0$, at iterations $k = 1, 2, \ldots$,

**1.** For all $1 \leqslant j \leqslant N$, agent $j$ samples $\xi_j^k$ chooses some $y_j^k \in \mathbb{R}^d$ as a $\mathcal{S}_{k-1}$-measurable function.

**2.** The shared memory is extended: $\mathcal{S}_k = \mathcal{S}_{k-1} \cup \left\{ g_j^k(y_j^k), \xi_j^k, 1 \leqslant j \leqslant N \right\}$.

**3.** Agent $j$ outputs $x_j^k$ as a $\mathcal{S}_k$-measurable function.

For fixed target precision $\varepsilon > 0$, the objective is to find, using $T_\varepsilon$ samples from all agents in total - corresponding to $K_\varepsilon = T_\varepsilon/N$ oracle calls -, models with local generalization error $\varepsilon$. Throughout the paper, we assume that each function $f_i$ is minimized over $\mathbb{R}^d$, and we denote by $x_i^\star$ such a minimizer. We further consider the following two standard assumptions.

**Assumption 1** (Noise). *There exists $\sigma^2 > 0$ such that for all $1 \leqslant i \leqslant N$ and $x \in \mathbb{R}^d$,*

$$\mathbb{E}_{\xi_i \sim \mathcal{D}_i} \|\nabla_x \ell(x, \xi_i) - \nabla f_i(x)\|^2 \leqslant \sigma^2 \,.$$

**Assumption 2** (Regularity). *Functions $f_i$ are $\mu$-strongly convex and $L$-smooth [7].*

## 2.2 Distribution-based distances

We first introduce extensions of classical *Integral Probability Metrics* (IPMs, [47]) to multivariate functions, i.e. pseudo-distances on the set of probability measures parameterized by a set $\mathcal{H}$ of functions, fixed in the sequel.

**Definition 1.** *For $\mathcal{H}$ a set of functions from $\Xi$ to $\mathbb{R}^d$ and $\mathcal{D}, \mathcal{D}'$ two probability distributions on $\Xi$, we define:*

$$d_\mathcal{H}(\mathcal{D}, \mathcal{D}') = \sup_{h \in \mathcal{H}} \|\mathbb{E}\left[h(\xi) - h(\xi')\right]\| \,,$$

*where $\xi \sim \mathcal{D}$ and $\xi' \sim \mathcal{D}'$. $d_\mathcal{H}$ is a pseudo-distance on the set of probability measures on $\Xi$.*

This family of pseudo-distances contains a large number of standard distances between distributions, including total variation (with the set of 1-locally bounded functions, functions that send any ball of radius 1 in a ball of radius 1), the Wassertein distance (with the set of 1-Lipschitz functions), maximum mean discrepancies (with the unit ball of a RKHS), or even a simple distance between means of the distributions (with the set of 1-Lipschitz affine functions), developed further in Section 6.

**Assumption 3** (Distribution-based dissimilarities). *For some non-negative weights $(b_{ij})_{1 \leqslant i,j \leqslant N}$, we have $d_\mathcal{H}(\mathcal{D}_i, \mathcal{D}_j) \leqslant b_{ij}$ for all $1 \leqslant i, j \leqslant N$. We further assume that either of the following holds.*

1. *(Weak dissimilarities). For all $1 \leqslant i \leqslant N$, $\left(\xi \in \Xi \mapsto \nabla_x \ell(x_i^\star, \xi)\right) \in \mathcal{H}$.*

2. *(Strong dissimilarities). For all $x \in \mathbb{R}^d$, $\left(\xi \in \Xi \mapsto \nabla_x \ell(x, \xi)\right) \in \mathcal{H}$.*

The "weak dissimilarities" assumption is of course easier to satisfy than the "strong" version, and our results will ultimately depend only on the weak assumption. Under Assumption 3 (weak version) and Assumption 2, we have $f_i(x_j^\star) - f_i(x_i^\star) \leqslant b_{ij}^2/(2\mu)$, which motivates our use of distribution-based dissimilarity assumptions.

**Notation:** in the rest of the paper, variables $t$ or $T$ denote the number of stochastic gradients $g_i^k$ sampled (or data item sampled from personal distribution) from all agents, while variables $k$ or $K$ denote the iterates of the algorithms (or equivalently the number of oracle calls made).

# 3 Information-theoretic lower bound on the sample complexity

In this section, we prove lower bounds on the total number of stochastic gradients required from all agents, to reach $\varepsilon$-generalization for a given agent. Our lower bounds apply to collaborative GEM, i.e. functions $(f_i)_{1 \leqslant i \leqslant N}$ of the form (1), for shared loss function $\ell$ and user distributions $\mathcal{D}_1, \ldots, \mathcal{D}_N$.

An oracle $\phi : \mathbb{R}^{N \times d} \to \mathcal{I}$ is a random function that answers some $\phi(x) \in \mathcal{I}$ where $\mathcal{I}$ is an information set, for every query $x \in \mathbb{R}^{N \times d}$. We adapt the definitions of Agarwal et al. [1] of sample complexity for *SGD* to our personalization problem. Formally, the *first-order* oracle we defined in Section 2 and that we write as $\phi\left((\mathcal{D}_i)_{i=1,\ldots,N}, \ell\right)$ for shared loss function $\ell$ and user distributions $\mathcal{D}_1, \ldots, \mathcal{D}_N$, returns for $x \in \mathbb{R}^{N \times d}$:

$$\phi\left((\mathcal{D}_i)_i, \ell\right)(x) = \left(i, \, x_i, \, \xi_i, \, \ell(x_i, \xi_i), \, g_i^k(x_i)\right)_{1 \leqslant i \leqslant N},$$

where $\xi_i \sim \mathcal{D}_i$. Given distributions and a loss function $\left((\mathcal{D}_i)_i, \ell\right)$, we denote by $\mathbb{M}$ the set of all methods $\mathcal{M} = (\mathcal{M}_K)_{K \geqslant 0}$: for any $K \geqslant 0$, $\mathcal{M}_K$ makes $K$ oracle calls from oracle $\phi\left((\mathcal{D}_i)_i, \ell\right)$ (while using $T = NK$ stochastic gradient samples from all agents), and returns $x_i^K \in \mathbb{R}^d$ for agent

$i$ as a measurable function of the $K$ oracle calls. For a set $\mathbb{D}$ of couples of distributions and loss function $\big((\mathcal{D}_j)_j, \ell\big)$ defining functions $(f_i)_{1 \leqslant i \leqslant N}$, we are interested in lower-bounding:

$$\inf_{\mathcal{M} \in \mathbb{M}} \sup_{((\mathcal{D}_j)_j, \ell) \in \mathbb{D}} \mathcal{K}_i^\varepsilon\Big(\mathcal{M}, \big((\mathcal{D}_j)_j, \ell\big)\Big),$$

where $\mathcal{K}_i^\varepsilon\Big(\mathcal{M}, \big((\mathcal{D}_j)_j, \ell\big)\Big)$ is the number of oracle calls required to reach generalization error $\varepsilon > 0$ for agent $i$, and writes as:

$$\mathcal{K}_i^\varepsilon\Big(\mathcal{M}, \big((\mathcal{D}_j)_j, \ell\big)\Big) = \inf\left\{K \in \mathbb{N}^* \text{ such that } \mathbb{E}\left[f_i(x_i^K) - \min_{x \in \mathbb{R}^d} f_i(x)\right] \leqslant \varepsilon\right\}.$$

We now define the set $\mathbb{D}$ we consider for our lower bounds. Let $b = (b_{ij})_{1 \leqslant i,j \leqslant N}$ be non-negative weights that verify the triangle inequality – namely, $b_{ij} \leqslant b_{ik} + b_{kj}$ for all $i, j, k$ –, and let $r, \mu, L, \sigma > 0$. $\mathbb{D}_\mu^L(r, b, \sigma)$ is the set of all $\big((\mathcal{D}_i)_{1 \leqslant i \leqslant N}, \ell\big)$, such that the functions $f_i$ parameterized by these tuples of distributions and shared loss function verify Assumptions 1, 2 and 3 for $\sigma^2, \mu, L > 0$ and $b$, such that $\|x_i^\star\| \leqslant r$ for all $1 \leqslant i \leqslant N$, and such that $f_i(x_j^\star) - f_i(x_i^\star) \leqslant b_{ij}^2/(2\mu)$. We use the notation $a(\cdot) = \Omega(b(\cdot))$ for $\exists C > 0$ such that $a(\cdot) \geqslant Cb(\cdot)$.

**Theorem 1** (IT lower bound). *Let $\varepsilon \in (0, 1/16)$, $(b_{ij})$ verifying the triangle inequality, $r, \sigma > 0$. Assume that the function set $\mathcal{H}$ contains the all 1-Lipschitz affine functions and that $d_\mathcal{H} \leqslant d_\mathrm{TV}$. For any $i \in \{1, \dots, N\}$:*

$$\inf_{\mathcal{M} \in \mathbb{M}} \sup_{((\mathcal{D}_j)_j, \ell) \in \mathbb{D}_{\mu=1/r^2}^{L=1/r^2}(r, b, \sigma)} \mathcal{K}_i^\varepsilon\Big(\mathcal{M}, \big((\mathcal{D}_j)_j, \ell\big)\Big) = \Omega\left(\frac{r^2\sigma^2}{\varepsilon} \times \frac{1}{\mathcal{N}_i^\varepsilon(\frac{b^2}{4\mu})}\right),$$

*where $\mathcal{N}_i^\varepsilon(\frac{b^2}{4\mu}) = \sum_j \mathbb{1}_{\{b_{ij}^2 \leqslant 4\mu\varepsilon\}}$ is the number of agents $j$ verifying $b_{ij}^2 \leqslant \varepsilon$.*

The proof of this lower bound (Appendix B) builds on lower bounds based on Fano's inequality [21] for stochastic gradient descent [1] or for information limited statistical estimation [20, 55], adapted to personalization. Theorem 1 states that, given the knowledge of $(b_{ij})$, $\sigma^2$, $\mu = 1$ and $L = 1$, there exist difficult instances of the problem that satisfy all three Assumptions 1, 2 and 3, such that the number of oracle calls needed to obtain a generalization error of $\varepsilon$ for an agent $i$ is lower-bounded by the right hand side of the equation in Theorem 1.

The factor $C\sigma^2 r^2 \varepsilon^{-1}$ is reminiscent of stochastic gradient descent, and is present in Agarwal et al. [1]: without cooperation, this is the sample complexity of *SGD* for a fixed agent. Cooperation appears in the factor $1/\mathcal{N}_i^\varepsilon(b/4)$: the sample complexity is inversely proportional to the number of agents $j$ that have distributions similar to that of $i$. One cannot hope for better than a linear *collaboration speedup* proportional to agents $4\mu\varepsilon$-close to $i$ in terms of the distance $d_\mathcal{H}$. Theorem 1 is a *worst-case* lower bound, so that a collaboration speedup could be leveraged even for small $\varepsilon$, but this would require making stronger additional assumptions.

## 4 The ALL-FOR-ONE algorithm: parallel weighted gradient averagings

After providing lower complexity bounds in Theorem 1, we present in this section a naive algorithmic approach based on weighted gradient averagings (WGA), that proves to be sample-optimal. Each agent $i$ keeps $N$ shared local models $x_1^k, \dots, x_N^k$, where $x_j^k$ estimates $x_j^\star$ at iteration $k$ (the knowledge of $x_j^k$ needs to be shared by all agents). At each iteration $k$, when a sample $\xi_j^k$ is obtained at agent $j$, it is used by that agent to compute unbiased estimates of $\nabla f_j(x_i^k)$ for all $i \in [N]$. The iterates of the WGA algorithm write as, where $\lambda_{ij} \geqslant 0$ are such that $\sum_j \lambda_{ij} = 1$ for all $1 \leqslant i \leqslant N$:

$$x_i^{k+1} = x_i^k - \eta \sum_{j=1}^N \lambda_{ij} g_j^k(x_i^k), \tag{2}$$

for some step size $\eta > 0$. We call this algorithm that consists in performing $N$ parallel WGA algorithms ALL-FOR-ONE (AFO), since every iteration of each gradient averaging for a given node $i$ requires all the other nodes to compute one stochastic gradient for $i$. WGA is thus equivalent to training models on the mixture of distributions $(\mathcal{D}_j)_j$ with weights $(\lambda_{ij})_j$ for all $i$.

**Theorem 2.** *Let $(x_i^k)_{1 \leqslant i \leqslant N, \, k \geqslant 0}$ be generated with* (2)*, and assume that Assumptions 1, 2 and 3 (strong version) hold. For any $K \geqslant 0$ and $1 \leqslant i \leqslant N$, and for $\eta$ as in Equation* (6)*,*[1]

$$\mathbb{E}\left[f_i(x_i^K) - f_i(x_i^\star)\right] \leqslant \left(f_i(x_i^{(0)}) - f_i(x_i^\star)\right)e^{-\frac{K}{2\kappa}} + \tilde{\mathcal{O}}\left(\frac{\kappa\sigma^2}{\mu K}\sum_{1 \leqslant j \leqslant N}\lambda_{ij}^2\right) + \sum_{1 \leqslant j \leqslant N}\lambda_{ij}\frac{b_{ij}^2}{\mu}\,.$$

*Let $\varepsilon > 0$. For a specific choice of $\lambda_{ij} = \frac{\mathbb{1}_{\left\{b_{ij}^2 < \varepsilon/2\right\}}}{\mathcal{N}_i^\varepsilon(2b)}$, WGA* (2) *satisfies* $\mathbb{E}\left[f_i(x_i^{K_\varepsilon(i)}) - f_i(x_i^\star)\right] \leqslant \varepsilon$ *for a number of oracle calls of:*

$$K_i(\varepsilon) = \tilde{\mathcal{O}}\left(\frac{\kappa\sigma^2}{\mu\varepsilon}\frac{1}{\mathcal{N}_i^\varepsilon(2b^2/\mu)}\right)\,,$$

*where $\mathcal{N}_i^\varepsilon$ is previously defined in Theorem 1.*

Since the oracle complexity of the WGA algorithm matches that of our lower bound, this proves that our lower bound is optimal. However, this algorithm may be difficult to use in practice: *(i)* the choice of $\lambda_{ij}$ is an explicit function of distribution distances $(b_{ij})$ (defined in Assumption 3) that can be (statistically speaking) as hard to compute as solving our optimization problem; and *(ii)* the memory requirements and computation/communication costs of WGA can be prohibitive for large $N$ and large $\varepsilon$ (they scale with $\mathcal{N}_i^\varepsilon$ for agent $i$). Note that the strong version of Assumption 3 used in Theorem 2 can be replaced by a more classical uniform bound of the form $\|\nabla f_i - \nabla f_j\| \leqslant b_{ij}$.

We first begin by solving this latter issue – an algorithmic one – in the next section, by introducing and studying the ALL-FOR-ALL algorithm. We discuss *(i)* in Section 6, where we provide scenarii over which statistical theoretical guarantees can be derived on the error made by estimating these distribution distances using only a few samples.

## 5   The ALL-FOR-ALL algorithm

---
**Algorithm 1** *All-for-all* algorithm
---
1:  Step size $\eta > 0$, matrix $W \in \mathbb{R}^{N \times N}$, initialization $x_1^0 = \ldots = x_N^0 \in \mathbb{R}^d$ ($x_i^0$ at agent $i$).
2:  **for** $k = 0, 1, 2, \ldots K - 1$ **do**
3:      Agents $1 \leqslant j \leqslant N$ compute $g_j^k(x_j^k)$ and broadcast it to all agents $i$ such that $W_{ij} > 0$.
4:      For $i = 1, \ldots, N$, update:

$$x_i^{k+1} = x_i^k - \eta\sum_{j\,:\,W_{ij}>0}W_{ij}g_j^k(x_j^k)\,.$$

5:  **end for**    Return $x_i^K$ for agent $i$

---

In this section, we present the ALL-FOR-ALL algorithm (AFA), an adaptation of the weighted gradient averaging algorithm. For $1 \leqslant i \leqslant N$, initialize $x_i^0 = x_0 \in \mathbb{R}^d$. At iteration $k$, let $x_i^k \in \mathbb{R}^d$ be agent $i$'s current estimate of $x_i^\star$, and denote $x^k = (x_i^k)_{1 \leqslant i \leqslant N} \in \mathbb{R}^{N \times d}$. For a step size $\eta > 0$ and a matrix $W \in \mathbb{R}^{N \times N}$ with non-negative entries (remarkably and as discussed later, $W$ will not necessarily verify $\sum_j W_{ij} = 1$), iterates of the *all-for-all* algorithm are generated with Algorithm 1. In Theorem 3, we control the averaged local generalization error amongst all agents:

$$F^k = \frac{1}{N}\sum_{i=1}^N f_i(x_i^k) - f_i(x_i^\star)\,, \quad k \geqslant 0\,.$$

**Theorem 3** (ALL-FOR-ALL algorithm)**.** *Let $K > 0$, $\eta > 0$, and $W$ a matrix of the form $W = \Lambda\Lambda^\top$ for some stochastic matrix $\Lambda = (\lambda_{ij})_{1 \leqslant i, j \leqslant N}$. Assume that Assumptions 1, 2 and 3 (weak version) hold. The iterates $(x_i^k)_{k \geqslant 0, 1 \leqslant i \leqslant N}$ generated with Algorithm 1 verify, for $\eta$ as in Equation* (7)*:*

$$\mathbb{E}\left[F^K\right] \leqslant F^0 e^{-\frac{K}{2\kappa}} + \tilde{\mathcal{O}}\left(\frac{\kappa\sigma^2}{K\mu N}\sum_{1 \leqslant i, j \leqslant N}\lambda_{ij}^2\right) + \frac{1}{N}\sum_{1 \leqslant i, j \leqslant N}\lambda_{ij}\frac{b_{ij}^2}{2\mu}\,.$$

---
[1] $\tilde{\mathcal{O}}$ hides logarithmic and constant factors

As for the AFO algorithm, we can deduce from this result the number of oracle calls required by the ALL-FOR-ALL algorithm to reach an averaged $\varepsilon$-generalization, under the idealistic setting where the distribution-based distances $b_{ij}$ are accessible.

**Corollary 1.** *Let $\varepsilon > 0$. Under the same assumptions as in Theorem 3, for a choice of matrix $W = \Lambda\Lambda^\top$ where $\lambda_{ij} = \frac{\mathbb{1}_{\left\{b_{ij}^2/\mu < \varepsilon\right\}}}{\mathcal{N}_i^\varepsilon(b^2/\mu)}$, the ALL-FOR-ALL algorithm (Algorithm 1) returns $(x_i^K)_{1 \leqslant i \leqslant N}$ satisfying $\frac{1}{N}\sum_{i=1}^N f_i(x_i^{K_\varepsilon}) - f_i(x_i^\star) \leqslant \varepsilon$, for a number of oracle calls satisfying:*

$$K_\varepsilon \leqslant 2 \max\left( \frac{\kappa\sigma^2}{\varepsilon\mu} \frac{1}{N}\sum_{i=1}^N \frac{1}{\mathcal{N}_i^\varepsilon(b^2/\mu)}, \; \kappa \right) \ln\left(\varepsilon^{-1} F^0\right).$$

Denoting $K_\varepsilon(i)$ the oracle complexity of the WGA algorithm - that matches the lower bound -, we observe that the ALL-FOR-ALL algorithm reaches an averaged $\varepsilon$-generalization with an number of oracle calls $K_\varepsilon$, of $K_\varepsilon \leqslant \frac{1}{N}\sum_i K_\varepsilon(i)$. The speedup in comparison with a no-collaboration strategy (all agents locally performing SGD) is $\frac{1}{N}\sum_i \frac{1}{\mathcal{N}_i^\varepsilon(b^2/\mu)}$: the mean of all local speedups.

**Remark 1.** *In Theorem 3, as its proof shows, the quantities $b_{ij}^2/(2\mu)$ in the last term can in fact be replaced by the quantities $f_i(x_j^\star) - f_i(x_i^\star)$, that control how well the optimal model for $j$ generalizes for $i$, and the bias induced by ALL-FOR-ALL iterations is a weighted average of these quantities. Note that in our lower bound (Theorem 1), we enforce that the functions considered are required to satisfy $f_i(x_j^\star) - f_i(x_i^\star) \leqslant b_{ij}^2/(2\mu)$. We believe this notion of function proximity that we leverage to be the weakest achievable in our setting; no prior work uses such a mild proximity assumption.*

Perhaps surprisingly, matrix $W$ is in general *not* a gossip matrix (*i.e.* such that $W\mathbb{1} = \mathbb{1}$): agent $i$ does not aggregate a convex combination of stochastic gradients, but a combination with scalars that do not necessarily sum to 1. We thus cannot say that the ALL-FOR-ALL algorithm acts as if, in parallel, each agent $i$ trains a model on the mixture of distributions $\mathcal{D}_j$ with weights $W_{ij}$. In fact, as the analysis shows below, agent $i$ trains a model on the mixture of distributions, with weights $\lambda_{ij}$, if $\Lambda$ is a stochastic square root of matrix $W$ ($\Lambda\Lambda^\top = W$), as in the AFO algorithm. In order to account for inter-dependencies between agents that do not directly share information, the *all-for-all gradient filtering* uses weights $W_{ij}$ to aggregate information, instead of $\lambda_{ij}$. Propagating information using a matrix $W$, that induces a similarity graph $G_W$ on $\{1, \ldots, N\}$, such that $(ij) \in E_W$ if $W_{ij} > 0$, is quite natural [4, 50]; yet, ours is the first analysis to give such precise generalization error bounds, through the use of a stochastic optimization framework.

In comparison to Theorem 3, the classical personalized FL approaches that consider personalized local models of the form $x_i = \bar{x} - \delta_i$, where $\bar{x}$ is some global quantity shared by all agents, perturbed (and personalized) by some local quantity $\delta_i$ (*e.g.* averaging between local and a global models), can be seen as the special instances where, for all $i$, we have $\lambda_{ii} = 1 - \alpha_i$ and $\lambda_{ij} = \frac{\alpha_i}{N-1}$ if $i \neq j$ for some $\alpha_i$, and leads to bias terms of the form $\frac{1}{N}\sum_i \frac{\alpha_i}{N-1}\sum_{j \neq i} b_{ij}$ [14, 25, 39]. Full and naive collaboration (a single model trained for all users) corresponds to $\lambda_{ij} = 1/N$ for all $i, j$, and leads to a bias term of $\frac{1}{N^2}\sum_{i,j} b_{ij}$. The degrees of freedom offered by our matrix $W$ (and by coefficients $\lambda_{ij}$) enable pairwise agent adaptation, and tighter generalization guarantees and bias/variance tradeoffs.

*Proof sketch of Theorem 3.* Since brutally analyzing convergence of the iterates $(x^k)$ generated with $x^{k+1} = x^k - WG^k$ seems impossible due to both gradient biases and model biases between agents, we study these iterates through the introduction of a different but related problem. This approach is in fact similar to some decentralized optimization ones, where a dual problem or a related energy function is often introduced [23, 45], upon which well-studied algorithms are applied. The related problem we formulate is different from and more flexible than all the different personalized FL problems in the literature [28, 48], that consider regularization terms that enforce consensus. For $\lambda = (\lambda_{ij})_{1 \leqslant i,j \leqslant N}$ a stochastic matrix (such that for all $1 \leqslant i \leqslant N$, we have $\sum_{j=1}^N \lambda_{ij} = 1$), let $f^\Lambda$ be defined as:

$$f^\Lambda(y) = \bar{f}(\Lambda y), \quad y \in \mathbb{R}^{N \times d}, \tag{3}$$

where $\bar{f} = \frac{1}{N}\sum_i f_i$. Gradient descent on $f^\Lambda$ writes as $y^{k+1} = y^k - \eta\Lambda^\top\nabla\bar{f}(\Lambda y^k)$, where $\nabla\bar{f}(x) = \frac{1}{N}\left(\nabla f_i(x_i)\right)_{1 \leqslant i \leqslant N}$ for any $x \in \mathbb{R}^{N \times d}$. Importantly, notice that denoting $x^k = \Lambda y^k$ and since $W = \Lambda\Lambda^\top$, we have the recursion $x^{k+1} = x^k - \eta W\nabla\bar{f}(x^k)$, making an analysis of the iterates

$(x^k)$ possible. In our case, we however use stochastic gradients given by our oracle. The full gradient $\nabla \bar{f}(x)$ is thus replaced by $(g_i^k(x_i))_i$. Defining $(y^k)_k$ with the recursion: $y^{k+1} = y_k - \eta((\Lambda G^k(y^k)_i)_i$, initialized at $y_1^0 = x_1^0 = \ldots = y_N^0 = x_N^0$ we have $x^k = \Lambda y^k$, for all $k \geqslant 0$, where $(x^k)$ is generated using Algorithm 1. As a consequence, controlling in Theorem 3 the function values $\frac{1}{N} \sum_i f_i(x_i^k)$ is equivalent to controlling $f^\Lambda(y^k)$ (these two quantities are equal). The bias-variance trade-off thus writes as, where $y^\Lambda$ minimizes $f^\Lambda$ and $x^\star = (x_i^\star)_{1 \leqslant i \leqslant N}$:

$$F^k \leqslant \underbrace{f^\Lambda(y^\Lambda) - \bar{f}(x^\star)}_{\text{Bias term}} + \underbrace{f^\Lambda(y^k) - f^\Lambda(y^\Lambda)}_{\text{Optimization and variance terms}} \quad .$$

The rest of the proof, deferred to Appendix D, consists in relating these two terms to $\Lambda$ and $b$. $\qquad \square$

After providing the optimization tools and results to answer for the shortcomings of weighted gradient averagings, we now turn to quantifying the impact of the use of estimated values $\hat{b}_{ij}$ of $b_{ij}$, in order to close the loop.

## 6  Estimation of $d_{\mathcal{H}}(\mathcal{D}_i, \mathcal{D}_j)$ as a pre-processing step

The sample complexity of estimating the distances $d_{\mathcal{H}}(\mathcal{D}_i, \mathcal{D}_j)$ depends on the complexity of the function space $\mathcal{H}$. While the estimation of Wasserstein or total variation distances are usually hard (in $\mathcal{O}(1/S^{1/d})$ where $d$ is the ambient dimension and $S$ the number of samples available for the estimation, see *e.g.* [52]), maximum mean discrepancy (MMD) distances often exhibit lower sample complexities in $O(1/\sqrt{S})$ [47]. Moreover, explicit assumptions on the loss function can also provide low sample complexities, as shown below for quadratic loss functions. Yet, the results presented in this section can be generalized beyond linear models with squared losses, as long as concentration inequalities for controlling how far empirical distributions are from the true distribution in terms of distance $d_{\mathcal{H}}$.

In order to formulate statistical results for the estimation of the pairwise distribution-based distances $d_{\mathcal{H}}(\mathcal{D}_i, \mathcal{D}_j)$, we need to make additional structural assumptions, on both $\mathcal{H}$ and the distributions. Inspired by Collins et al. [11], we focus on analyzing an instance of our general GEM setting for quadratic losses and linear models, under which the generalization error of a given agent $i$ writes as:

$$f_i(x) = \frac{1}{2} \mathbb{E}\left[ \left( a_i^\top x - b_i \right)^2 \right], \quad x \in \mathbb{R}^d,$$

where $z_i = (a_i, b_i)$ is a random variable on $\mathbb{R}^d \times \mathbb{R}$. The stochastic gradients thus write as $\nabla_x \ell(x, \xi_i) = (a_i^\top x - b_i)a_i$ for $z_i = (a_i, b_i)$, and are thus linear functions of $\xi_i = z_i z_i^\top$. Hence, Assumption 3 (weak version) is satisfied for $\mathcal{H}$ the set of $D^\star$-Lipschitz and affine functions, where $D^\star$ bounds all $\|x_i^\star\|$ for $1 \leqslant i \leqslant N$, leading to:

$$d_{\mathcal{H}}(\mathcal{D}_i, \mathcal{D}_j) \leqslant D^\star \|\mathbb{E}[\xi_i] - \mathbb{E}[\xi_j]\|.$$

We make the following assumption on the law $\mathcal{D}_i$ of the random variables $\xi_i$: they are non-isotropic subgaussian random variables, that thus benefit from concentration inequalities that are dimension-independent [22, 35].

**Assumption 4.** *For some non-negative symetric matrix $\Sigma$ and all $1 \leqslant i \leqslant N$, $\xi_i$ are centered and $\Sigma$-subgaussian:*

$$\mathbb{P}\left(\xi_i^\top y \geqslant u\right) \leqslant \exp\left(-\frac{u^2}{2y^\top \Sigma y}\right), \quad \forall y \in \mathbb{R}^{(d+1)^2}, \quad \forall u > 0,$$

*and we denote as $\nu^2$ the largest eigenvalue of $\Sigma$, and $d_{\text{eff}} = \frac{\|\Sigma\|_2}{\nu^2}$ its effective dimension.*

Importantly, note that $d_{\text{eff}}$ can be arbitrarily smaller than the ambient dimension - for the MNIST dataset, $d_{\text{eff}}$ is less than 3, while the ambient dimension is 712 [22]. Depending on a smaller dimension is also an assumption that Collins et al. [11] use in their work by exploiting shared representations.

We now formulate a structural assumption on the set of agents: there are $M$ clusters $\mathcal{C}_1, \ldots, \mathcal{C}_M$ of $C$ agents each (to ease notations, with a total number of agents $N = MC$). Within each cluster, agents distributions share the same objective, and clusters are "well-separated". These models are popular for modelling population heterogeneity and provide a formal framework for clustering problems; we refer the interested reader to Melnykov and Maitra [41] for a detailed survey on the subject.

**Assumption 5** (Well-separated clusters of agents). *For $M, C \geqslant 1$, $N$ writes as $N = MC$ and there exists $\{\mathcal{C}_1, \ldots, \mathcal{C}_M\}$ a partition of $\{1, \ldots, N\}$, $\mu_1, \ldots, \mu_m$ such that for all $1 \leqslant m \leqslant M$, $|\mathcal{C}_m| = C$ and for all $i, j \in \mathcal{C}_m$, we have $\mathbb{E}[\xi_i] = \mathbb{E}[\xi_j] = \mu_m$. We denote $\Delta^2 = \min_{m \neq m'} \|\mu_m - \mu_{m'}\|^2$ and assume that $\Delta^2 > 0$.*

When distribution-based distances were given (as in Corollary 1), Algorithm 1 achieved the optimal collaboration speedup, linear in $1/C$ under Assumption 5 and for small enough target precision $\varepsilon$. The cluster model is thus the natural baseline for our problem. In the case where agents estimate with whom to collaborate as we do in the sequel, reaching this collaboration speedup of $1/C$ will hence prove the effectiveness of the approach.

We assume that agents possess a limited number of samples. More precisely, for $1 \leqslant i \leqslant N$ and $S, K \geqslant 1$, agent $i$ possesses $K + S$ i.i.d. samples of drawn from $\mathcal{D}_i$, $S$ of which are dedicated to estimating who to collaborate with, the $K$ remaining dedicated to the optimization process *i.e.* to running ALL-FOR-ALL iterations for a number $K$ of oracle calls.

For $1 \leqslant i \leqslant N$, let $\hat{\mu}_i$ be an estimation of $\mathbb{E}[\xi_i]$ made with $S$ i.i.d. samples $\xi_{i,1}, \ldots, \xi_{i,S}$, and for $1 \leqslant i, j \leqslant N$ let $\hat{b}_{ij}$ be the following estimation of $d_{\mathcal{H}}(\mathcal{D}_i, \mathcal{D}_j)$:

$$\hat{\mu}_i = \frac{1}{S} \sum_{s=1}^{S} \xi_{i,s}, \quad \hat{b}_{ij} = \|\hat{\mu}_i - \hat{\mu}_j\|.$$

Computing these distances can be done using only $\tilde{\mathcal{O}}(N)$ communications (rather than the $N^2$ communications of a naive approach) by performing randomized gossip communications [6] on the complete graph.

**Theorem 4.** *Under Assumptions 1,2,3 and 5, and for $\varepsilon > 0$, the ALL-FOR-ALL algorithm with estimated biases as just described reaches an averaged generalization error of $\varepsilon$ as long as the number of pre-used samples $S$ and of oracle calls $K$ satisfy:*

$$S = \tilde{\Omega}\Big(\frac{\nu^2 d_{\text{eff}}}{\Delta^2}\Big), \quad C = \tilde{\Omega}\Big(\frac{\nu^2 d_{\text{eff}}}{\varepsilon}\Big), \quad KC = \tilde{\Omega}\Big(\frac{\kappa \sigma^2}{\varepsilon \mu}\Big), \quad K = \tilde{\Omega}(\kappa).$$

**Theorem 5** (ALL-FOR-ALL with estimated biases). *Assume that Assumptions 1, 2, 3 (for some unknown biases $b_{ij}$), 4 and 5 hold. Under the setting described, let $\hat{\Lambda}$ be the stochastic matrix with entries*

$$\hat{\lambda}_{ij} = \frac{\mathbb{1}_{\{\hat{b}_{ij}^2 \leqslant u\}}}{\sum_{\ell=1}^{N} \mathbb{1}_{\{\hat{b}_{i\ell}^2 \leqslant u\}}},$$

*for some $u > 0$ that verifies $u \geqslant \frac{4\nu^2 d_{\text{eff}}}{S}$. The ALL-FOR-ALL algorithm with $W = \hat{\Lambda}\hat{\Lambda}^\top$ outputs $(x_i^K)_{1 \leqslant i \leqslant N}$ verifying, where $b_{\max} = \max_{i,j} b_{ij}$:*

$$\mathbb{E}[F^K] \leqslant F^0 e^{-\frac{K}{2\kappa}} + \tilde{\mathcal{O}}\left(\frac{\kappa\sigma^2}{K\mu}\Big(\frac{1}{C} + Ce^{-\frac{Su}{8\nu^2}}\Big)\right) + \frac{2D^{\star 2} b_{\max} e^{-\frac{S\max(\Delta^2 - 2u, 2u^2)}{8\nu^2}}}{\mu} + 4u^2 \mathbb{1}_{\{2u \geqslant \Delta\}},$$

*where the mean is taken over both biases estimates $(\hat{b}_{ij})$ and gradient estimates $(g_i^k)$.*

**Corollary 2.** *Under the same assumptions as Theorem 5 and for $\varepsilon > 0$, the ALL-FOR-ALL algorithm with estimated biases as described above reaches an averaged generalization error of $\varepsilon$ as long as:*

$$S = \tilde{\Omega}\Big(\frac{\nu^2 d_{\text{eff}}}{\Delta^2}\Big), \quad C = \tilde{\Omega}\Big(\frac{\nu^2 d_{\text{eff}}}{\varepsilon}\Big), \quad KC = \tilde{\Omega}\Big(\frac{\kappa \sigma^2}{\varepsilon \mu}\Big), \quad K = \tilde{\Omega}(\kappa).$$

Forgetting about the logarithmic factors, only a bounded number of local samples for each user ($S$ and $K$) are required to reach an averaged arbitrarily small generalization error $\varepsilon > 0$, in the limit with an arbitrary large number of agents ($N$ and $C$). Indeed, due to our regularity assumptions, $K$ – the number of samples kept for the optimization problem – is required only to be of order $\kappa$, the condition number of the problem. The number of samples $S$ used for estimating the biases is required to be of order $\nu_1^2 d_{\text{eff}}/\Delta^2$, the "signal-to-noise" ratio of our mixture model [41, 46], a natural quantity to depend on. Corollary 2 hence shows that the optimal collaboration speedup is achieved, up to logarithmic factors: in order to reach an arbitrary small generalization error $\varepsilon > 0$, are only required

constant orders for $S$ and $K$ (the number of samples locally available) if the number of agents is large enough *i.e.* if $N = \tilde{\Omega}(M/\varepsilon)$, where $M$ is the number of clusters *i.e.* we have a linear speedup in the clusters population. We numerically illustrate our theory in Appendix A on synthetic datasets, with clustered agents (as in this section), as well as in a setting where agents are distributed according to a more general "distribution of agent".

A closely related work [26] also studies a model where agents verify a cluster structure as described in Assumption 5 for quadratic losses and linear models. Yet, we highlight several differences between their approach and ours. First, Ghosh et al. [26] perform an *online* clustering of the agents, as opposed to our pre-training hierarchical approach. While the results we obtain in Theorem 5 and Corollary 2 and those of Ghosh et al. [26] have the same linear speedup in the number of agents, ours require no initialization condition. Finally, our algorithm is decentralized, thus leading to improved scalability (especially in terms of the number of clusters) and privacy [13], if of interest. Finally, not being restricted to clusters in the analysis of the ALL-FOR-ALL algorithm leads to a better collaboration speedup and fairness (in the sense that performance does not impact a few agents) in a non-clustered scenario, where an approach based on clusters would be highly non-optimal for agents that are at the border of the inferred clusters.

## Conclusion

In this paper, we quantified in terms of function and distribution biases, stochastic gradient noise, target precision $\varepsilon > 0$ and functions regularity parameters, the benefit of collaboration between agents for shared minimization using stochastic gradient algorithms. Our lower bound (Theorem 1) states that, under prior knowledge on the distances between local distributions, the collaborative speedup can be linear only in the first phase of the optimization when the generalization error is large compared to the distances between distributions. More specifically, for a given agent $i$, the collaboration speedup is linear in the number of agents that are $\varepsilon$-close to $i$. Moreover, we show that the ALL-FOR-ONE algorithm allows such a speedup and is thus sample optimal. However, this algorithm requires high computation and communication capacities, a drawback that can be mitigated by the use of a novel algorithm called ALL-FOR-ALL, that benefits from the same collaboration speedup while being cheaper to deploy. Finally, we studied the impact of estimating distances between distributions as a pre-processing step to the optimization phase; under a mixture model assumptions on the agents, we obtain an optimal collaboration speedup. Extending our results – lower and upper complexity bounds – to other regularity assumptions and Section 6 to more general settings, as well as incorporating local steps in the ALL-FOR-ALL algorithm (even though to lighten communications, unbiased compressors could here be used, as our analysis encompasses these) are interesting questions left for future work. See [24] for extensions to convex-smooth functions (not necessarily strongly convex), convex-Lipschitz functions, and to asynchronous gradient oracles.

**Aknowledgments** This work was supported by ANR-19-P3IA-0001(PRAIRIE 3IA Institute). We also acknowledge support from the MSR-INRIA joint centre.

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
