## A Numerical illustration of our theory

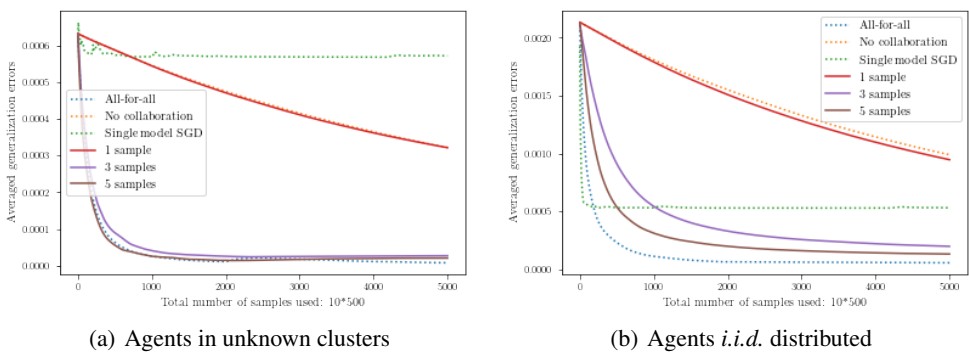

(a) Agents in unknown clusters

(b) Agents *i.i.d.* distributed

Figure 1: **All-for-all algorithm in practice**

To test the robustness of our theory, we build toy problems from synthetic datasets, placing ourselves in the scenario we considered throughout the paper: a large number of agents with heterogeneous data, that each have too few samples available from their local data distribution in order to reach a small generalization error on their own.

In Figure 1, we consider $N = 500$ agents, and a quadratic loss function $\ell(x, \xi = (a, b)) = \frac{1}{2}(a^\top x - b)^2$, for $x, a \in \mathbb{R}^d$ ($d = 100$) and $\mu \in \mathbb{R}^d$. For $i = 1, \ldots, 500$, the distribution $\mathcal{D}_i$ of $\xi_i = (a_i a_i^\top, a_i b_i)$ as a centered Gaussian random variable of covariance matrix $\Sigma_i$ for $a_i$, and $b_i$ is the sign of $a_i^\top u$ for some fixed $u \in \mathbb{R}^d$, flipped with probability 0.2. In both figures, each agents have 10 samples available for the optimization phase ($K = 10$ oracle calls), corresponding to a total number of samples used of $N \times K = 5000$. We computed and showed the 500 steps of all ten oracle calls, each step corresponding to the use of the stochastic gradient of a single agent.

The dotted lines represent our baselines. The blue one is the ALL-FOR-ALL algorithm with matrix $W$ exactly as in Corollary 1 with $b_{ij} = \|\Sigma_i - \Sigma_j\|$. The orange dotted line consists in the no-collaboration baseline: each agents performs SGD on its own without sharing information (corresponds to $W = I_N$). The green dotted line corresponds to the "single-model" approach without personalization: one model is trained for all agents, using SGD and all samples from all agents (corresponding to $W = \frac{1}{n} \mathbb{1}\mathbb{1}^\top$). The choice of the algorithm for the single model approach without collaboration is in fact unimportant, since all algorithms would reach the same asymptotic bias here. The full lines (red, violet and brown) correspond to estimating the pairwise distances from empirical distributions (as in Section 6), using respectively $S = 1$, $S = 3$ and $S = 5$ samples.

In Figure 1(a), we consider $M = 10$ (unkonwn) clusters $\mathcal{C}_1, \ldots, \mathcal{C}_M$. All $i \in \mathcal{C}_m$ have the same covariance matrix $\Sigma_m$, equal to $I_d/\sqrt{d} + e_m e_m^\top$, where $e_m$ is the $m$-th element of the canonical basis of $\mathbb{R}^d$. In Figure 1(b), $\Sigma_i = \mathrm{Diag}(u_1^{(i)}, \ldots, u_1^{(d)})/\sqrt{d}$ where the $(u_\ell^{(i)})$ are *i.i.d.* uniformly distributed in $[0, 1]$. Performing rough estimations of the pairwise distance between agents' local distributions thus appears to be quite robust in both our settings. In the "cluster" setting, this was predicted by our theory, and the numerical results are compelling. In the "*i.i.d.*" setting, using very few samples for the estimation also appears to be very efficient.

## B Proof of our lower-bound (Theorem 1)

### B.1 General framework to prove lower bounds [1]

The idea is that, when optimizing a function $f(x) = \mathbb{E}[\ell(x, \xi)]$ and finding a good approximation of a minimizer $x^\star$, we learn some information on the distribution $\mathcal{D}$ over which samples are drawn. In order to prove lower bounds, we construct a loss function $\ell$, and distributions $\mathcal{D}_1^\alpha, \ldots, \mathcal{D}_N^\alpha$, where $\alpha$ is a random parameter. We argue that minimizing the objective function up to a certain precision gives a good estimator (quantified) of the random seeds $\alpha$. Then, using Fano inequality, we bound

the efficiency of such an estimator in terms of number of oracle calls, obtaining a lower bound on the sample complexity. This approach is inspired by Agarwal et al. [1], who prove IT-lower bounds for stochastic gradient descent. We adapt their proof technique to the personalized and multi-agent setting.

**Constructing difficult loss functions** For any two functions $f, g : \mathbb{R}^d \to \mathbb{R}$, we define the discrepancy measure $\rho(f, g)$ as:

$$\rho(f, g) = \inf_{x \in \mathbb{R}^d} \left\{ f(x) + g(x) - \inf_{y \in \mathbb{R}^d} f(y) - \inf_{y \in \mathbb{R}^d} g(y) \right\},$$

which is a pseudo metrics. Now, for a finite set $\mathcal{V}$ of parameters, let $\mathcal{G}(\delta) = \left\{ g_\alpha^\delta, \, \alpha \in \mathcal{V} \right\}$ be a set of functions indexed by $\mathcal{V}$, that depend on $\delta$ (fixed in the set). The dependency in $\delta$ of each $g_\alpha \in \mathcal{G}(\delta)$ is left implicit in the following subsections. We define:

$$\psi(\delta) = \inf_{f, g \in \mathcal{G}(\delta), f \neq g} \rho(f, g).$$

**Minimizing is Bernoulli parameters identification** The two following lemmas justify that optimizing a function $g_\alpha \in \mathcal{G}(\delta)$ to a precision of order $\psi(\delta)$ is more difficult than estimating the parameter $\alpha$.

**Lemma 1** (Agarwal et al. [1]). *For any $x \in \mathbb{R}^d$, there can be at most one function $g_\alpha$ in $\mathcal{G}(\delta)$ such that:*

$$g_\alpha(x) - \inf_{\mathbb{R}^d} g_\alpha < \frac{\psi(\delta)}{3}.$$

**Lemma 2** (Agarwal et al. [1]). *Assume that for some fixed but unknown $\alpha \in \mathcal{V}$ there exists a method $\mathcal{M}_K$ based on the data $\phi = \{X_1, ..., X_K\}$ that returns $x^K$ (function of $\phi$) satisfying an error of:*

$$\mathbb{E}\left[ g_\alpha(x^K) - \min_{x \in \mathbb{R}^d} g_\alpha(x) \right] < \frac{\psi(\delta)}{9},$$

*where the mean is taken over the randomness of both the oracle $\Phi$, the method $\mathcal{M}_K$ and $\alpha \in \mathcal{V}$ if random. Then, there exists a hypothesis test $\hat{\alpha} : \phi \to \mathcal{V}$ such that:*

$$\max_{\alpha \in \mathcal{V}} \mathbb{P}_\phi\big(\hat{\alpha} \neq \alpha\big) \leqslant \frac{1}{3}.$$

Suppose now that the parameter $\alpha$ in the previous Lemma is chosen uniformly at random in $\mathcal{V}$. Let $\hat{\alpha} : \phi \to \mathcal{V}$ be a hypothesis test estimating $\alpha$. By Fano inequality [12], we have:

$$\mathbb{P}\big(\hat{\alpha} \neq \alpha\big) \geqslant 1 - \frac{I\big(\phi, \alpha\big) + \ln(2)}{\ln(|\mathcal{V}|)}, \tag{4}$$

where $I\big(\phi, \alpha\big)$ is the mutual information between $\phi$ and $\alpha$, that we need to upper-bound. Combining Fano inequality with Lemmas 1 and 2, fixing a target error $\varepsilon = \psi(\delta)$, we obtain a lower bound on the $K_\varepsilon$ the number of oracle calls required to reach an $\varepsilon$ generalization error:

$$\frac{1}{3} \geqslant \mathbb{P}_\phi\big(\hat{\alpha} \neq \alpha\big) \geqslant 1 - \frac{I\big(\phi_{K_\varepsilon}, \alpha\big) + \ln(2)}{\ln(|\mathcal{V}|)},$$

where $\phi_{K_\varepsilon}$ is the information contained in $K_\varepsilon$ oracle calls. If we have an equality of the form $I\big(\phi_{K_\varepsilon}, \alpha\big) = K_\varepsilon I\big(\phi_1, \alpha\big)$, this gives:

$$K_\varepsilon \geqslant \frac{\frac{2}{3} \ln(|\mathcal{V}|) - \ln(2)}{I(\phi_1, \alpha)}. \tag{5}$$

Playing with the different parameters $\delta, \alpha, \mathcal{V}$ gives lower bounds. We refer the interested reader to Chapter 2 in Cover and Thomas [12] for Fano inequality and mutual information.

## B.2 Applying this to prove Theorem 1

For simplicity, assume that $r^2 = d$ and $\sigma^2 = 1$. Let $\delta > 0$ a free parameter. Let $\mathcal{V} = \{\alpha^1, \ldots, \alpha^L\} \subset \{-1, 1\}^d$ be a subset of the hypercube such that for all $k \neq l$,

$$\frac{1}{2} \sum_{i=1}^d |\alpha_i^k - \alpha_i^l| \geqslant \frac{d}{4} \,,$$

*i.e.* $\mathcal{V}$ is a $d/4$-packing of the hypercube. We know that we can set $|\mathcal{V}| \geqslant (2/\sqrt{e})^{d/2}$. Without loss of generality, we prove a lower bound in the case where the agent that desires to minimize its local function is indexed by 1.

Let:

$$\ell(x, \xi) = \frac{1}{2} \|x - \xi\|^2 \,,$$

for $x, \xi \in \mathbb{R}^d$ and, for fixed $\delta > 0$ and any $\alpha \in \mathcal{V}$:

$$g_\alpha(x) = \frac{1}{2d} \sum_{k=1}^d \left( x_k^2 + 1 - 2(\frac{1}{2} + \alpha_k \delta) x_k \right) \quad , x \in \mathcal{X} \,.$$

We keep the same notations as last subsection ($\psi(\delta), \rho$). We have:

$$\rho(g_\alpha, g_\beta) = \frac{\delta^2}{d} \sum_{k=1}^d |\alpha_k - \beta_k| \,,$$

leading to $\psi(\delta) \geqslant \delta^2/4$ since $\mathcal{V}$ is a $d/4$-packing of the hypercube.

For any $i = 1, \ldots, N$, let $\mathcal{D}_i$ be the probability distribution on $\{0, 1\}^d$ of the following random variable:

$$\mathrm{Ber}\left(\frac{1}{2} + \delta_i \alpha_k\right) \epsilon_k \quad \text{where} \quad \delta_i = (\delta - b_{i1})^+ \,,$$

where $s^+ = \max(0, s)$ for $s \in \mathbb{R}$, $k$ is taken uniformly at random in $\{1, \ldots, d\}$, $(\epsilon_k)$ is the canonical basis of $\mathbb{R}^d$, and $Ber(p)$ is a Bernoulli random variable, independent of $k$.

The mutual information is thus, in our case:

$$I(\phi_K, \alpha) \leqslant C_1 K \mathcal{N}_1^\delta(\sqrt{b}) \delta^2 \,,$$

where we use the fact that $I\left(\mathrm{Ber}\left(\frac{1}{2} + \mathbb{1}_{b_{1i} \leqslant \delta} \alpha_k \delta_i\right), \alpha_k\right) \leqslant C_1 \delta^2$ for some constant $C_1 > 0$, for $\delta_i \leqslant 1/4$. Setting the target precision as $\varepsilon = \delta^2/4$, we obtain:

$$K_\varepsilon \geqslant C' \frac{d}{\varepsilon \mathcal{N}_1^\varepsilon(4b)} \,.$$

The loss function and distributions built verify our regularity assumptions for $\mu = 1/d$, $L = 1/d$, noise $\sigma^2 \leqslant 1$.

We first verify that for all $1 \leqslant j, k \leqslant N$, we have $f_j(x_k^\star) - f_j(x_j^\star) \leqslant b_{kj}^2$. We first notice that $x_j^\star = \frac{1}{d}\left(\frac{1}{2} + \delta_j \alpha_l\right)_{1 \leqslant l \leqslant d}$, so that:

$$
\begin{aligned}
f_j(x_k^\star) - f_j(x_j^\star) &= \frac{1}{d} \|x_j^\star - x_k^\star\|^2 \\
&= (\delta_i - \delta_k)^2 \\
&\leqslant (b_{1j} - b_{1k})^2 \\
&\leqslant |b_{1j} - b_{1k}|^2 \\
&\leqslant b_{jk}^2 \,,
\end{aligned}
$$

since the weights $b$ verify the triangle inequality. Under the assumptions of Theorem 1 on $\mathcal{H}$, we have, in terms of distribution-based distances:

$$d_\mathcal{H}(\mathcal{D}_i, \mathcal{D}_j) \leqslant |\delta_i - \delta_j| \leqslant b_{ij} \,.$$

The minimum of each $g_\alpha$ is attained at $x^\alpha = \frac{1}{2} + \delta\alpha$, we thus need to assume that $r$ is of order $\sqrt{d}$, and a rescaling leads to the dependency in $r$. The dependency in $\sigma^2$ for $\sigma^2 > 1$ is obtained by taking $\mathcal{D}_i' = \mathrm{Ber}(1/\sigma^2)\sigma^2 \mathcal{D}_i$. In this case, we have a noise amplitude of order $\sigma^2$ instead of order 1, and a factor $1/\sigma^2$ appears in the mutual information.

## C   Proof of Theorem 2

*Proof of Theorem 2.* We begin by proving the following descent lemma.

**Lemma 3.** *Let $F : \mathbb{R}^d \to \mathbb{R}$ be $L$-smooth and $\mu$-strongly convex and $G : \mathbb{R}^d \to \mathbb{R}$ be differentiable. Consider the iterates generated by:*

$$y^{k+1} = y^k - \eta g^k \,,$$

*where $\mathbb{E}\left[g_k|y^k\right] = \nabla G(y^k)$ and $\mathbb{E}\left[\|g^k - \nabla G(y^k)\|^2|y^k\right] \leqslant \sigma_g^2$. Then, we have, where $y^\star$ minimizes $F$, as long as $\eta \leqslant 1/L$:*

$$\mathbb{E}\left[F(y^{k+1}) - F(y^\star)\right] \leqslant (1 - \eta\mu)\mathbb{E}\left[F(y^k) - F(y^\star)\right] + \frac{\eta}{2}\mathbb{E}\left[\|\nabla F(y^k) - \nabla G(y^k)\|^2\right] + \frac{\eta^2 \sigma_g^2 L}{2}$$

*Proof Lemma 3.* We use smoothness of $F$:

$$\mathbb{E}\left[F(y^{k+1}) - F(y^k)\right] \leqslant -\eta\mathbb{E}\left[\langle g^k, \nabla F(y^k)\rangle\right] + \frac{\eta^2 L}{2}\mathbb{E}\left[\|g^k\|^2\right] \,.$$

Then, using $\mathbb{E}\left[\|g^k\|^2\right] \leqslant \mathbb{E}\left[\|\nabla G(y^k)\|^2\right] + \sigma_g^2$, and $-\eta\mathbb{E}\left[\langle g^k, \nabla F(y^k)\rangle\right] = -\eta\mathbb{E}\left[\langle \nabla G(y^k), \nabla F(y^k)\rangle\right] = -\frac{\eta}{2}\mathbb{E}\left[\|\nabla G(y^k)\|^2 + \|\nabla F(y^k)^2\| - \|\nabla G(y^k) - \nabla F(y^k)\|^2\right]$, we obtain that:

$$\mathbb{E}\left[F(y^{k+1}) - F(y^k)\right] \leqslant -\frac{\eta}{2}\mathbb{E}\left[\|\nabla F(y^k)\|^2\right] - \frac{\eta}{2}\mathbb{E}\left[\|\nabla G(y^k)\|^2\right]\left(1 - \eta L\right)$$

$$+ \frac{\eta}{2}\mathbb{E}\left[\|\nabla F(y^k) - \nabla G(y^k)\|^2\right] + \frac{\sigma_g^2 \eta^2 L}{2} \,.$$

Finally, we conclude using $-\frac{\eta}{2}\mathbb{E}\left[\|\nabla F(y^k)\|^2\right] \leqslant \eta\mathbb{E}\left[F(y^k) - F(y^\star)\right]$ and $\eta < 1/L$.   □

Let $i \in [N]$. To prove Theorem 2, we now use Lemma 3 to study the sequence $y^k = x_i^k$ with $F = f_i$, $G = \sum_j \lambda_{ij}f_j := f^\lambda$ and $g^k = \sum_j \lambda_{ij}g_j^k(x_i^k)$. For all $x \in \mathbb{R}^d$, using Assumption 3:

$$\left\|\nabla f_i(x) - \nabla f^\lambda(x)\right\|^2 \leqslant \sum_{j=1}^N \lambda_{ij}\|f_i(x) - f_j(x)\|^2$$

$$= \sum_{j=1}^N \lambda_{ij}\|\mathbb{E}\left[\nabla_x \ell(x, \xi_i)\right] - \mathbb{E}\left[\nabla_x \ell(x, \xi_j)\right]\|^2$$

$$\leqslant \sum_{j=1}^N \lambda_{ij}b_{ij}^2 \,,$$

since $\nabla_x \ell(x, \cdot) \in \mathcal{H}$ and $d_{\mathcal{H}}(\mathcal{D}_i, \mathcal{D}_j) \leqslant b_{ij}$. Then, using the independence (conditionally on $x_i^k$) of the $(g_j^k)_j$ and $\mathbb{E}\left[g_j^k(x_i^k)|x_i^k\right] = \nabla f_j(x_i^k)$:

$$\mathbb{E}\left[\left\|\sum_j \lambda_{ij}g_j^k(x_i^k) - \sum_j \lambda_{ij}\nabla f_j(x_j^k)\right\|^2\right] = \sum_j \mathbb{E}\left[\|\lambda_{ij}(g_j^k - \nabla f_j(x_j^k))\|^2\right] \leqslant \sum_j \lambda_{ij}^2 \sigma^2 \,.$$

Consequently,

$$\mathbb{E}\left[f_i(x_i^{k+1}) - f_i(x_i^\star)\right] \leqslant (1 - \eta\mu)\mathbb{E}\left[f_i(x_i^k) - f_i(x_i^\star)\right] + \frac{\eta}{2}\sum_{j=1}^N \lambda_{ij}b_{ij}^2 + \frac{\eta^2 \sigma^2 L}{2}\sum_{j=1}^N \lambda_{ij}^2 \,.$$

Writing $H^k = (1 - \eta\mu)^{-k}\mathbb{E}\left[f_i(x_i^k) - f_i(x_i^\star)\right]$, unrolling the recursion leads to:

$$H^K \leqslant H^0 + \frac{\eta}{2}\sum_{k<K}(1 - \eta\mu)^{-k}\sum_{j=1}^N \lambda_{ij}b_{ij}^2 + \sum_{k<K}(1 - \eta\mu)^{-k}\frac{\eta^2 \sigma^2 L}{2}\sum_{j=1}^N \lambda_{ij}^2 \,.$$

Finally, using $\sum_{k<K}(1-\eta\mu)^{-k} \leqslant \frac{(1-\eta\mu)^{-K}}{\eta\mu}$, we have:

$$\mathbb{E}\left[f(x_i^K) - f(x_i^\star)\right] \leqslant (1-\eta\mu)^K(f(x_i^0) - f(x_i^\star)) + \frac{\eta\sigma^2 L}{2\mu}\sum_{j=1}^N \lambda_{ij}^2 + \frac{1}{2\mu}\sum_{j=1}^N \lambda_{ij}b_{ij}^2.$$

Using $(1-\eta\mu)^K \leqslant e^{-K\eta\mu}$, we optimize of $\eta$. For

$$\eta_i = \min\left\{\frac{1}{2L}, \frac{1}{\mu K}\ln\left(\frac{2\mu^2 K(f(x_i^0) - f(x_i^\star))}{\sigma^2 L \sum_j \lambda_{ij}^2}\right)\right\}, \tag{6}$$

we obtain:

$$\mathbb{E}\left[f(x_i^K) - f(x_i^\star)\right] \leqslant (f(x_i^0) - f(x_i^\star))e^{-K/\kappa}$$
$$+ \frac{\sigma^2 L}{\mu^2 K}\ln\left(\frac{2\mu^2 K(f(x_i^0) - f(x_i^\star))}{\sigma^2 L \sum_j \lambda_{ij}^2}\right)\sum_j \lambda_{ij}^2 + \frac{1}{2\mu}\sum_{j=1}^N \lambda_{ij}b_{ij}^2,$$

leading to the first part of Theorem 2. For the second part, we simply plug the expression of $\lambda_{ij}$ in the proven formula. $\square$

## D    Proof of Theorem 3

We recall that for a stochastic matrix $\Lambda$, we defined

$$f^\Lambda(y) = \frac{1}{N}\sum_{i=1}^N f_i\left(\sum_{j=1}^N \lambda_{ij}y_j\right), \quad y = (y_1, \ldots, y_N) \in \mathbb{R}^{N\times d}.$$

Then, $y^\Lambda$ is defined as a minimizer of $f^\Lambda$, and we write $x^\star = (x_1^\star, \ldots, x_N^\star)$ where $x_i^\star$ is the minimizer of $f_i$.

We first begin with the following simple lemmas.

**Lemma 4.** *If Assumption 3 (weak version) holds, then for all $i, j = 1, \ldots, N$, we have:*

$$f_i(x_j^\star) - f_i(x_i^\star) \leqslant \frac{b_{ij}^2}{2\mu}.$$

*Proof.* Using strong-convexity of $f_i$ and $\nabla f_i(x_i^\star)$:

$$f_i(x_j^\star) - f_i(x_i^\star) \leqslant \frac{1}{2\mu}\left\|\nabla f_i(x_j^\star)\right\|^2$$
$$= \frac{1}{2\mu}\left\|\nabla f_i(x_j^\star) - \nabla f_i(x_i^\star)\right\|^2$$
$$= \frac{1}{2\mu}\left\|\mathbb{E}\left[\nabla_x \ell(x_j^\star, \xi_i)\right] - \mathbb{E}\left[\nabla_x \ell(x_j^\star, \xi_i)\right]\right\|^2$$
$$\leqslant \frac{b_{ij}^2}{2\mu},$$

where the last inequality is deduced using the weak version of Assumption 3. $\square$

**Lemma 5.** *If $\Lambda$ is a stochastic matrix,*

$$f^\Lambda(y^\Lambda) - \bar{f}(x^\star) \leqslant \frac{1}{N}\sum_{1\leqslant i,j\leqslant N} \lambda_{ij}\left(f_i(x_j^\star) - f_i(x_i^\star)\right).$$

*Proof.* Writing the optimality of $y^\Lambda$ gives:

$$f^\Lambda(y^\Lambda) \leqslant f^\Lambda(x^\star)$$
$$= \frac{1}{N}\sum_i f_i\left(\sum_j \lambda_{ij}x_j^\star\right)$$
$$\leqslant \frac{1}{N}\sum_{1\leqslant i,j\leqslant N} \lambda_{ij}f_i(x_j^\star),$$

where we used convexity of each $f_i$. Then, subtracting $\bar{f}(x^\star)$ and using stochasticity of $\Lambda$:

$$f^\Lambda(y^\Lambda) - \bar{f}(x^\star) \leqslant \frac{1}{N} \sum_{1 \leqslant i,j \leqslant N} \lambda_{ij}(f_i(x_j^\star) - f_i(x_i^\star)).$$

$\square$

We are now armed to prove Theorem 3.

*Proof.* We have the following bias-variance decomposition, where the inequality is a consequence of Lemma 5:

$$\begin{aligned}
F^k &= \bar{f}(y^k) - \bar{f}(x^\star) \\
&= f^\Lambda(y^k) - f^\Lambda(y^\Lambda) + f^\Lambda(y^\Lambda) - \bar{f}(x^\star) \\
&\leqslant f^\Lambda(y^k) - f^\Lambda(y^\Lambda) + \frac{1}{N} \sum_{1 \leqslant i,j \leqslant N} \lambda_{ij} \frac{b_{ij}^2}{2\mu}.
\end{aligned}$$

We thus need to upper-bound the optimization term $f^\Lambda(y^k) - f^\Lambda(y^\Lambda)$. We recall that $y^k$ verifies the recursion:

$$y^{k+1} = y^k - \eta \nabla G_\Lambda^k(y^k),$$

for

$$G_\Lambda^k(y) = \frac{1}{N} \big( \sum_{i=1}^N \lambda_{ij} g_i^k((\Lambda y^k)_i)) \big)_{1 \leqslant j \leqslant N},$$

that verifies:

$$\begin{aligned}
\mathbb{E}\left[G_\Lambda^k(y)\right] &= \nabla f^\Lambda(y), \\
\mathbb{E}\left[\left\|G_\Lambda^k(y) - \nabla f^\lambda(y)\right\|^2\right] &\leqslant \frac{\sigma^2}{N^2} \sum_{1 \leqslant i,j \leqslant N} \lambda_{ij}^2.
\end{aligned}$$

The function $f^\Lambda$ is however not necessarily strongly convex. However, since $\nabla^2 f^\Lambda(y) = \Lambda^\top \nabla^2 \bar{f}(\Lambda y)\Lambda$ and $\bar{f}$ is $L/N$-smooth and $\mu/N$-strongly convex, $f^\Lambda$ is $L/N$-relatively smooth and $\mu/N$-relatively strongly convex [2] with respect to $\frac{1}{2}\|y\|_W^2 = \frac{1}{2}y^\top W y$. Note also that the spectral radius of $W$ is 1, since $\Lambda$ is stochastic. Instead of using stochastic Bregman gradient descent (*e.g.* Dragomir et al. [18]), we use Lemma 6 that we prove at the end of the paper: classical SGD that naturally generalizes to relative smoothness and strong convexity assumptions, when the mirror map is quadratic. This leads to:

$$\mathbb{E}\left[f^\Lambda(y^k) - f^\Lambda(y^\Lambda)\right] \leqslant (f^\Lambda(y^0) - f^\Lambda(y^\Lambda))e^{-\frac{K}{2\kappa}} + \frac{\kappa\sigma^2}{K\mu N} \ln\left(\frac{2\mu^2 K(f^\Lambda(y^0) - f^\Lambda(y^\Lambda))}{\sigma^2 L \sum_{i,j} \frac{1}{N}\lambda_{ij}^2}\right) \sum_{1 \leqslant i,j \leqslant N} \lambda_{ij}^2,$$

for a choice of stepsizes of:

$$\eta = \min\left\{\frac{1}{2L}, \frac{1}{\mu K} \ln\left(\frac{2\mu^2 K(f^\Lambda(y^0) - f^\Lambda(y^\Lambda))}{\sigma^2 L \sum_{i,j} \frac{1}{N}\lambda_{ij}^2}\right)\right\}, \tag{7}$$

concluding the proof. $\square$

# E   Proof of Theorem 5

We first start by recalling that, for a $\Sigma$-subgaussian random variable $\xi$, using Theorem 1 from [29], we have for any $t \geqslant 0$:

$$\mathbb{P}\left(\|\xi\|^2 \geqslant d_{\text{eff}}\nu^2 + 2\sqrt{\|\Sigma\|_2 t} + 2\nu^2 t\right) \leqslant e^{-t}.$$

Consequently, for $u \geqslant \max(\nu^2 d_{\text{eff}}, \|\Sigma\|_2^2/\nu^2)$, we have:

$$\mathbb{P}\left(\|\xi\|^2 \geqslant 4u\right) \leqslant e^{-\frac{u}{2\nu^2}} .$$

Remarking that $\frac{\|\Sigma\|_2^2}{\nu^2} = \nu^2 \sum_k \frac{\nu_k^4}{\nu^4} \leqslant \nu^2 \sum_k \frac{\nu_k^2}{\nu^2} = \nu^2 d_{\text{eff}}$, where $\nu_k^2$ are the eigenvalues of $\Sigma$, this condition on $u$ is in fact $u \geqslant \nu^2 d_{\text{eff}}$.

*Proof of our Theorem.* From Theorem 3, we have, conditionally on the $S$ samples used in estimating biases:

$$\mathbb{E}\left[F^K|\hat{\Lambda}\right] \leqslant F^0 e^{-\frac{K}{2\kappa}} + \tilde{\mathcal{O}}\left(\frac{\kappa\sigma^2}{K\mu N} \sum_{1 \leqslant i,j \leqslant N} \hat{\lambda}_{ij}^2\right) + \frac{1}{N} \sum_{1 \leqslant i,j \leqslant N} \hat{\lambda}_{ij} \frac{b_{ij}^2}{2\mu} .$$

We hence need to bound $\mathbb{E}\left[\sum_{i,j} \hat{\lambda}_{ij}^2\right]$ and $\mathbb{E}\left[\sum_{i,j} \hat{\lambda}_{ij} b_{ij}^2\right]$, and start by assuming that $u \geqslant \frac{4}{S}\nu^2 d_{\text{eff}}$.

First, denoting $b_{\max} = \max_{i,j} b_{ij}$, we have:

$$\begin{aligned}
\mathbb{E}\left[\sum_{i,j} \hat{\lambda}_{ij} b_{ij}^2\right] &= \sum_{i,j} \mathbb{E}\left[\hat{\lambda}_{ij} b_{ij}^2\right] \\
&= \sum_{i,j:\, b_{ij}^2 > 4u} \mathbb{E}\left[\hat{\lambda}_{ij} b_{ij}^2\right] + \sum_{i,j:\, b_{ij}^2 \leqslant 4u} \mathbb{E}\left[\hat{\lambda}_{ij} b_{ij}^2\right] \\
&\leqslant b_{\max}^2 \mathbb{P}\left(\hat{b}_{ij}^2 \leqslant u | b_{ij}^2 > 4u\right) + 4u^2 \mathbb{1}_{\{4u^2 \geqslant \Delta\}} .
\end{aligned}$$

Using a triangle inequality, that gives us $2\|\hat{\mu}_i - \hat{\mu}_j - \mathbb{E}[\hat{\mu}_i - \hat{\mu}_j]\|^2 \geqslant b_{ij}^2 - 2\hat{b}_{ij}^2$, $\mathbb{P}\left(\hat{b}_{ij}^2 \leqslant u | b_{ij}^2 > 4u\right) \leqslant \mathbb{P}\left(\|\hat{\mu}_i - \hat{\mu}_j - \mathbb{E}[\hat{\mu}_i - \hat{\mu}_j]\|^2 \geqslant \frac{b_{ij}^2 - 2u}{2}\right)$. Then, since each $\xi_i$ and $\xi_j$ are $\Sigma$-subgaussian (and independent), $\hat{\mu}_i - \hat{\mu}_j - \mathbb{E}[\hat{\mu}_i - \hat{\mu}_j]$ is $4\Sigma/S$ subgaussian[2], so that using our assumption on $u$, we can use Theorem 1 of Hsu et al. [29]:

$$\mathbb{P}\left(\hat{b}_{ij}^2 \leqslant u | b_{ij}^2 > 4u\right) = \mathbb{P}\left(\hat{b}_{ij}^2 \leqslant u | b_{ij}^2 > \max(4u, \Delta^2)\right) \leqslant 2e^{-\frac{S \max(2u, \Delta^2 - 2u)}{8\nu^2}} .$$

Then, for $1 \leqslant i \leqslant N$, let $\hat{\mathcal{N}}_i = \sum_{j=1}^{N} \mathbb{1}_{\{\hat{b}_{ij} \leqslant u\}}$. Fix $1 \leqslant m \leqslant M$. We have:

$$\begin{aligned}
\mathbb{P}\left(\forall i \in \mathcal{C}_m, \|\hat{\mu}_i - \mu_m\|^2 \leqslant u\right) &= 1 - \mathbb{P}\left(\exists i \in \mathcal{C}_m, \|\hat{\mu}_i - \mu_m\|^2 > u\right) \\
&\geqslant 1 - 2Ce^{-\frac{Su}{8\nu^2}} ,
\end{aligned}$$

so that $\mathbb{E}\left[\sum_{i \in \mathcal{C}_M} \sum_{1 \leqslant j \leqslant N} \lambda_{ij}^2\right] = \mathbb{E}\left[\frac{1}{C} \sum_{i \in \mathcal{C}_m} \frac{1}{\hat{\mathcal{N}}_i}\right] \leqslant \frac{1}{C} + 2Ce^{-\frac{Su}{8\nu^2}}$, concluding the proof. We then prove the resulting corollary by taking $u = \Delta/4$, and the condition on $u$ translates into $S \geqslant 16\frac{\nu^2 d_{\text{eff}}}{\Delta^2}$.

$\square$

# F    SGD under strong-convexity and smoothness assumptions

We recall the following well-known result.

---

[2]indeed, using the subgaussian norm $\|\cdot\|_{\psi_2}$, for real-valued independent random variables $X_1, \ldots, X_S$ and any $\beta > 0$, $\mathbb{E}\left[e^{\beta \frac{1}{S} \sum_{s=1}^{S} X_s}\right] = \prod_s \mathbb{E}\left[e^{\frac{\beta}{S} X_s}\right] \leqslant \prod_s \mathbb{E}\left[e^{C_{\psi_2} \frac{\beta^2}{S^2} \|X_s\|_{\psi_2}}\right] = \mathbb{E}\left[e^{C_{\psi_2} \frac{\beta^2}{S^2} \sum_{s=1}^{S} \|X_s\|_{\psi_2}}\right]$, so that $\left\|\frac{1}{S} \sum_{s=1}^{S} X_s\right\|_{\psi_2} \leqslant \frac{1}{S^2} \sum_{s=1}^{S} \|X_s\|_{\psi_2}$, and we apply this to the random variables $\hat{\mu}_i \top y$

**Lemma 6** (SGD, s.c. and smooth). *Define $\|x\|_A^2 = x^\top A x$ for some non-negative and symmetric matrix A. Let $f : \mathcal{X} \to \mathbb{R}$ $\mu$-relatively strongly convex and L-relatively smooth with respect to $\frac{1}{2}\|x\|_A^2$. Let $(f_t, g_t)_{t \geqslant 0}$ be first order oracle calls such that for all $t \geqslant 0$:*

$$\forall x \in \mathcal{X}, \quad \begin{cases} \mathbb{E}\left[f_t(x)\right] = f(x), \\ \mathbb{E}\left[g_t(x)\right] = \nabla f(x), \\ \mathbb{E}\left[\|g_t(x) - \nabla f(x)\|^2\right] \leqslant \sigma^2, \end{cases}$$

*for some $\sigma > 0$. Let $L_A$ be the largest eigenvalue of A, and assume that $L_A \leqslant 1$ (our result generalizes to any $L_A$). Let $(x_t)_{t \geqslant 0}$ be generated with:*

$$\forall t \geqslant 0, \quad x^{t+1} = x^t - \eta g_t(x^t),$$

*for a fixed stepsize $\frac{1}{2L} \geqslant \eta > 0$, and assume that all the iterates lie in $\mathcal{X}$. Assume that $f$ is minimized over $\mathcal{X}$ at some interior point $x^\star$. We have for any $T > 0$:*

$$\mathbb{E}\left[f(x^T) - f(x^\star)\right] \leqslant e^{-\eta \mu T}\left(f(x^0) - f(x^\star)\right) + \frac{\eta L \sigma^2}{\mu}.$$

*For fixed $T > 0$, setting $\eta = \min\left(1/(2L), \frac{1}{\mu T}\ln(\frac{f_0 \mu^2 T}{L\sigma^2})\right)$ gives:*

$$\mathbb{E}\left[f(x^T) - f(x^\star)\right] \leqslant e^{-\frac{\mu}{2L}T}\left(f(x^0) - f(x^\star)\right) + \frac{L\sigma^2}{\mu^2 T}\ln\left(\frac{f_0 \mu^2 T}{L\sigma^2}\right).$$

*Thus, for fixed target precision $\varepsilon > 0$, using stepsize $\eta_\varepsilon = \min\left(\frac{\mu\varepsilon}{2L\sigma^2}, \frac{1}{2L}\right)$ and setting $T_\varepsilon = \lceil \ln\left(\varepsilon^{-1}(f(x^0) - f(x^\star))\right)\frac{1}{\eta_\varepsilon \mu}\rceil$, we have:*

$$f\left(\frac{1}{T_\varepsilon}\sum_{t < T_\varepsilon} x^t\right) - f(x^\star) \leqslant \varepsilon,$$

*with a number of oracle calls*

$$T_\varepsilon \leqslant \max\left(\frac{2L\sigma^2}{\varepsilon\mu^2}, \frac{2L}{\mu}\right)\ln\left(\varepsilon^{-1}(f(x^0) - f(x^\star))\right).$$

*Proof.* For some $t \geqslant 0$, denoting $f_t = \mathbb{E}\left[f(x^{t+1}) - f(x^\star)\right]$, using relative smoothness, unbiasedness of the stochastic gradients and then relative strong convexity:

$$f_{t+1} - f_t \leqslant -\eta \mathbb{E}\left[\|\nabla f(x^t)\|^2\right] + \frac{\eta^2 L}{2}\mathbb{E}\left[\|g_t\|_A^2\right]$$

$$\leqslant -\eta \mathbb{E}\left[\|\nabla f(x^t)\|^2\right] + \frac{\eta^2 L L_A}{2}\mathbb{E}\left[\|g_t\|^2\right]$$

$$\leqslant -\eta\left(1 - \frac{\eta L L_A}{2}\right)\mathbb{E}\left[\|\nabla f(x^t)\|^2\right] + \frac{\eta^2 L L_A \sigma^2}{2}.$$

Using relative strong convexity of $f$, we have:

$$\|\nabla f(x^t)\|^2 \geqslant \frac{1}{L_A}\|\nabla f(x^t)\|_A^2$$

$$\leqslant \frac{2\mu}{L_A}f_t,$$

yielding, for $\eta < 1/(L L_A)$

$$f_{t+1} - f_t \leqslant -2\eta\frac{\mu}{L_A}f_t + \frac{\eta^2 L L_A \sigma^2}{2}.$$

Then, for some $T > 0$ and since $L_A \leqslant 1$, sum the above inequality multiplied by $(1 - \eta\mu)^{-t-1}$:

$$\sum_{0 \leqslant t \leqslant T-1}(1 - \eta\mu)^{-t-1}f_{t+1} - (1 - \eta\mu)^{-t}f_t \leqslant \frac{\eta^2 L \sigma^2}{2}\sum_{0 \leqslant t \leqslant T-1}(1 - \eta\mu)^{-t-1}$$

$$\leqslant \frac{\eta^2 L \sigma^2}{2}\frac{(1 - \eta\mu)^{-t-1}}{\eta\mu},$$

leading to the desired result. $\qquad\square$