# OpenReview forum: "On Sample Optimality in Personalized Collaborative and Federated Learning"
_NeurIPS.cc/2022/Conference — NeurIPS 2022 Accept_

### Official Review · Reviewer_WfLi · 2022-07-09

**Rating:** 6
**Confidence:** 2
**Soundness:** 4 excellent
**Presentation:** 4 excellent
**Contribution:** 4 excellent

**Summary:**

The paper shows an information-theoretic lower bound on the number of oracle calls for collaborative training on clients. Two optimal collaborative training algorithms (gradient filtering algorithms) are proposed, and convergence rates are presented.

**Questions:**

The paper in several sections is out of my area of expertise. My questions are as follows:

1. Theorem 2: given $\lambda_{ij} \geq 0$ and they sum up to 1, the 3rd term in the right-hand side of the inequality is lower bounded by $a_i := min_j \frac{b_{ij}^2}{\mu}$. The first two terms are at least zero. Does the theorem allow $\varepsilon$ to be less than $a_i$?

2. Why should the matrix $W$ in the equation between lines 53 and 54 be symmetric?

**Limitations:**

Limited experiments.

**Strengths And Weaknesses:**

Strengths. The paper develops a strong theory proving a lower bound on the sample complexity and proposing algorithms with matching upper bounds.

Weaknesses. The limited experimental support of the theory.

---

> ### Author Response · Authors · 2022-07-29
> **Response to Reviewer WfLi**
>
> We thank the reviewer for her/his positive feedback, and answer her/his questions below.
>
> - Answer to question 1: in Theorem 1, since $b_{ii}=0$ (agent $i$'s local distribution is at distance 0 from itself), $a_i=0$ in reviewer's Question 1, so that $\varepsilon$ can be taken arbitrarily small (not less than $a_i=0$, since $\varepsilon$ is the target error).
> However, if for all $j\ne i$, we have $b_{ij}>0$, and defining $a_i'=\min_{j\ne i} b_{ij}$, even though $\varepsilon$ is allowed to be less than $a_i'$, the algorithm will not benefit from a collaborative speedup for such a small $\varepsilon$: this is expected in light of the fundamental limit expressed in Theorem 1.
>
> - Answer to question 2: matrix $W$ between lines 53 and 54 is symmetric, in light of Theorem 3, where $W$ is required to write as $W=\Lambda\Lambda^\top$.

---

> > ### Comment · Reviewer_WfLi · 2022-08-07
> > **Thanks for your clarifications.**
> >
> > Thanks for your clarifications.

---

### Official Review · Reviewer_teu2 · 2022-07-10

**Rating:** 7
**Confidence:** 3
**Soundness:** 3 good
**Presentation:** 3 good
**Contribution:** 3 good

**Summary:**

This paper studies the issue of personalization in federated learning - a very timely topic. While several works have introduced a variety of techniques (e.g., clustering, representation learning, MAML, etc.,) to promote personalization in FL, this paper takes a different approach. It explores the idea of aggregating gradients by carefully adjusting the weights of such aggregation based on pairwise client similarities. An information-theoretic lower bound is established that reveals how much a client participating in the FL process can hope to benefit from collaboration. Furthermore, it is shown that the weighted aggregation scheme can achieve such a lower bound.



**Questions:**

I don't have any questions beyond the clarifying comments I raised earlier.

**Limitations:**

I don't foresee any negative societal impacts of this work.

**Strengths And Weaknesses:**

Overall, I think this is a very well-written paper. I summarize my main comments below.

*Strengths*
- The main contribution of the paper, in my opinion, is Theorem 1. It is very informative in that it reveals how much a client can reduce its local generalization error by using the data from other "similar" clients. In this context, the notions of similarity introduced in Assumption 3 are both novel and intuitive.

-  The All-For-All algorithm achieves the lower bound in Theorem 1 by employing a filtering protocol - while updating its model via gradient aggregation, a client only uses the gradients of other clients that are close to it in a suitable sense. As a result, the weight matrix $W$ used in the aggregation rule is no longer a standard Markov matrix used for studying consensus/distributed optimization. This makes the analysis of the algorithm a bit different from the standard analyses.

One key point to note here is that the algorithm assumes prior knowledge of the client similarity indices, which can be very prohibitive in practice. Fortunately, in Section 6, the authors show how one can use a small number of samples initially to figure out such similarity indices and accordingly cluster the agents.

Coupled with the fundamental limit in Theorem 1, the All-For-All algorithm and the initial similarity-index estimation protocol in Section 6 lead to a strong theoretical contribution. I do have a few clarifying comments that are outlined below.

*Comments*

- Other than being a conceptual tool, is there any practical utility of the All-For-One Algorithm? In particular, in addition to the stringent communication overhead it imposes, it seems that at each round, every client needs to evaluate the gradient of its loss function at the models of *all* clients. This is in stark contrast to existing FL algorithms, where every client computes gradients at just its own model/iterate. As such, the All-For-One algorithm seems to require significantly more oracle calls than standard FL algorithms. Is this a fair assessment?

- The approach outlined in Section 6 seems to be valid for simple quadratic loss functions. This is fine for now as it lends some credibility to the overall approach proposed in this paper. But is there hope that the results in Section 6 can be generalized beyond quadratic losses?

- At the end of the day, the overall technique boils down to clustering clients initially, and running a parallel gradient descent algorithm within each cluster; the benefits of collaboration are then limited to each cluster. In this sense, I am curious to know how this high-level idea compares with some other well-established clustering-based methods in FL. See, for instance, ref [R1] below.

[R0] *An Efficient Framework for Clustered Federated Learning*, Ghosh et al.., NeurIPS 20.

In [R0], the authors develop a strategy that involves alternatively refining the cluster identities and solving the supervised learning problem. That is, the clustering and learning processes are interleaved, as opposed to the hierarchical approach suggested in the current work. I would like the authors to comment on this further.

- The aspect of personalization stems from the fact that the data-generating distributions of the clients are non-identical in practice, leading to loss functions with different minima (indeed, if all clients saw data from identical distributions, the global model would also serve well as a personalized model). As such, a very rich body of work has explored the angle of objective/statistical heterogeneity, improving our understanding of this fundamental challenge in the context of FL. However, the authors in this paper completely ignore this line of work, which I find unsatisfactory. I would like to see a better representation (and discussion) of this literature in the authors' related work section. Some notable papers in this regard are mentioned below, but this is by no means an exhaustive list.

[R1] *Tighter Theory for Local SGD on Identical and Heterogeneous Data*, Mischenko et al., AISTATS 20.

[R2] *SCAFFOLD: Stochastic Controlled Averaging for Federated Learning*, Karimireddy et al., ICML 20.

[R3] *Minibatch vs local sgd for heterogeneous distributed learning*, Woodworth et al., NeurIPS 20.

[R4] *FedSplit: An algorithmic framework for fast federated optimization*, Pathak and Wainwright, NeurIPS 20.

[R5] *Convergence and accuracy trade-offs in federated learning and meta-learning*, Charles and Konecny, AISTATS 21.

[R6] *Federated Learning Based on Dynamic Regularization*, Acar et al., ICLR 21.

[R7] *Linear Convergence in Federated Learning: Tackling Client Heterogeneity and Sparse Gradients*, Mitra et al., NeurIPS 21.

[R8] *Local SGD: Unified Theory and New Efficient Methods*, Gorbunov et al., AISTATS 21.

---

> ### Author Response · Authors · 2022-07-29
> **Response to Reviewer teu2**
>
> We would like to thank the reviewer for the time spent for her/his thorough review of our paper, all the suggestions it contain, and the questions ask (answered below).
>
> - The all-for-one algorithm: other than when only one agen desires to minimize its local generalization error, the all-for-one algorithm indeed suffers from highcosts, and is thus more a conceptual tool (that proves that the lower bound of Theorem 1 is indeed sharp, and introduces and motivates the all-for-all algorithm) than an efficient FL algorithm, a limitation we are completely transparent with in our paper (aknowledged after Theorem 2).
> Depending on how oracle calls are defined (1 oracle call = 1 computation of a stochastic gradient, or = 1 sample from local distribution are 2 different ways to count oracle calls), the all-for-all algorithm may thus require significantly more oracle calls (of about an additional factor of $\max_i\mathcal{N}_i^\varepsilon(b)$), as righteously noted.
>
> - Generalization of Section 6 beyond quadratic functions: there is hope for a generalization of the results given in Section 6, due to the generality of our assumptions. As long as concentration inequalities exist for controlling how far the distance $d_{\mathcal{H}}$ between two empirical distributions taken with $S$ samples is far from the distance between the two true distributions,  our analysis can be performed. As explained at the beginning of Section 6, Wasserstein and TV distances are such distances, but require $\delta^d$ samples to reach a precision $\delta$ on this deviation, and can be hard to compute. Yet, these cover a whole spectrum of problems.
> If $\mathcal{H}$ is a RKHS (a fairly general case), the associated distances are MMDs (maximum mean discrepancies), that can have much favorable sample complexity, while computing the distances between empirical measures is as easier as in the quadratic case (the difference between the embeddings of the distributions in the RKHS). Another way to generalize Section 6 is to relax the cluster assumption, and rather assume that agents are drawn from a distribution of agents in an \emph{i.i.d.} fashion. Depending on the structure of the ‘‘distribution of agents'' (for instance, the cluster assumption boils down to taking a mixture of $C$ components for this distribution), our analysis may apply.
>
> - Comparison with [R0] An Efficient Framework for Clustered Federated Learning, Ghosh et al.., NeurIPS 20.:
> The approach of [R0] is very similar to ours in Section 6, except that as noted, their clustering is performed in an online fashion, as opposed to our pre-training hierarchical approach. The results we obtain in the cluster+quadratics setting seem to recover the results of [R0]  with less restrictive initialization conditions. We highlight the following differences between [R0] and our approach: (i) [R0]'s algorithm and analysis is specifically designed to handle agents that are clustered, while our approach is more general and has the possibility to be generalized to many situations (ii) our algorithm is decentralized (no central entity required), as opposed to [R0], leading to better scalability (and privacy, if of interest), in particular to the number of clusters (that should remained bounded in [R0] to avoid communication, computation and memory costs to rise); (iii) not being restricted to clusters in the analysis of the all-for-all algorithms leads to a better collaboration speedup and fairness (in the sense that performance does not impact a few agents) in a non-clustered scenario, where  an approach based on clusters would be highly non-optimal for agents that are at the border of the inferred clusters.
>
> - Comparison to the [R1-R8] line of work (and other related works): references suggested tackle data-heterogeneity in distributed learning (by studying Local SGD and variants): data-heterogeneity is known to slow down the optimization when using Local SGD/ FedAvg  ([R1] e.g.) that trains 1 model for all agents, and many works studied algorithms that overcame this downside (e.g., scaffold). These works are related to our paper in the sense that data-heterogeneity is studied. However, these works study the training of \textbf{one} model (under heterogeneous data) for all agents, and are thus orthogonal to personalization (even though a very recent and interesting paper -ref. [A] below- studies local SGD as a means to learn local representations for personalization).
> Due to the impact of this line of work, we acknowledge our mistake not to have discussed these in the related works section, which will do so in a revised version, in order to put in perspective data-heterogeneity in distributed learning as a challenge to design fast and scalable federated optimization algorithms, with data-heterogeneity as a challenge for statistical meaning of the model (or models, in our case) trained.
>
> [A] FedAvg with Fine Tuning: Local Updates Lead to Representation Learning, Collins et al. 2022.

---

> > ### Comment · Reviewer_teu2 · 2022-08-07
> > **Thanks for the Response**
> >
> > Thank you for commenting on the points raised by me during the review. I am happy with the responses, and look forward to the revised manuscript with the updates that the authors have promised to make. I have no further comments, and would like to recommend acceptance.

---

### Official Review · Reviewer_g22N · 2022-07-12

**Rating:** 6
**Confidence:** 4
**Soundness:** 3 good
**Presentation:** 2 fair
**Contribution:** 3 good

**Summary:**

This paper studies the sample complexity of personalized federated learning with a large number of users (each holding a few samples). The authors investigate the benefit of collaborative learning via stochastic gradient descent in terms of distribution biases, noise variance, and regularity of function classes (strongly convex and smooth optimization). Based on known distances between local distributions, the authors propose a lower bound on the sample complexity of all users in order to minimize the generalization error of one user. They also present the All-For-One algorithm, which achieves the linear speedup shown in the lower bound and propose a second modified algorithm to address the communication and computation costs associated with this method.

**Questions:**

.

**Limitations:**

.

**Strengths And Weaknesses:**

The mathematics in this paper appear to be correct, and the topic is interesting. The organization and writing of this paper are generally very good. Moreover, the authors have provided an experimental result to support their theoretical findings. A limitation of this study is that it does not compare with seminal work on personalized federated learning. A comparison of the two proposed algorithms (All-For-One and All-For-All) has not been conducted with other techniques for personalization.

---

> ### Author Response · Authors · 2022-07-29
> **Response to Reviewer g22N**
>
> We thank the reviewer for the overall positive feedback and for providing a direction to make our arguments more convincing. Since our paper is mostly theoretical, experiments in Appendix A aimed at illustrating the theory, and we believe that validating our theory through experiments goes beyond the scope of our paper, since it would require running experiments on various datasets and compare with different benchmarks.
> However, we propose to add more detailed comparisons in our paper after our convergence results (in particular after Theorem 3 and Corollary 1) between our rates of convergence (number of communications, reliance on a central coordinator, number of local computations and most importantly to us and the focus of this paper, the number of samples required) and the rates of the references [13,22,33] (we are open to any additional suggestion regarding this list) furnished in our paper. We propose also to compare extensively to the reference [R0] of Reviewer teu2 (see details in the discussion with Reviewer teu2).
> In particular, [22] doesn't obtain $\varepsilon$-generalization error for arbitrary small $\varepsilon$. [33] only considers distances between the mixture of all distributions and local distributions, and therefore obtains (in all 3 approaches proposed) a bias that cannot vanish.
> The same phenomenon happens for [13].

---

### Official Review · Reviewer_tnmx · 2022-07-13

**Rating:** 6
**Confidence:** 2
**Soundness:** 2 fair
**Presentation:** 3 good
**Contribution:** 2 fair

**Summary:**

The paper studies the personalized federated learning and makes a few contributions. First, it proves an information-theoretic lower bound on the sample complexity. Second, it proposes an all-for-one algorithm that matches the lower bound but needs a high communication cost for broadcasting stochastic gradients between agents. Third, it further proposes a communication-efficient all-for-all algorithm via the gradient-filtering method.

**Questions:**

In Line 179, how does the factor epsilon/4 come?

In Line 203, large epsilon?

**Limitations:**

see above

**Strengths And Weaknesses:**

originality: theoretical contributions to FL. It leverages the distribution-based distance to analyze the sample complexity.

quality: technical depth looks sufficient.

clarity: good except that it is unclear how the analysis differs from existing works that is closest to this work.

significance: gradient filtering seems to be a good technique for both communication reduction and personalization promotion.

---

> ### Author Response · Authors · 2022-07-29
> **Response to Reviewer tnmx**
>
> We thank the reviewer for her/his positive feedback and review. We answer the two questions raised:
>
> - ‘‘In Line 179, how does the factor epsilon/4 come?'':
> There is a typo in our paper, which we apologize for: it is a factor $4\mu\varepsilon$ rather than $\varepsilon/4$: we modified this in the newly updated and posted version  of our paper (alongside with other minor typos we spotted between the submission date and the rebuttal).
> This factor comes from Theorem 1, just above line 179: the linear collaborative speedup cannot outperform the number of agents that verify
> $b_{ij}^2\leq 4\mu \varepsilon$ (due to $\mathcal{N}_i^\varepsilon (b^2/(4\mu))$ in the RHS of Theorem 1, where having $N$ instead would mean full collaboration speedup) , hence this factor.
>
> - ‘‘In Line 203, large epsilon?'': for large $N$, quantities $\mathcal{N}_i^\varepsilon$ increase. This is also true as $\varepsilon$ increases, even though this case might not be the most interesting, since $\varepsilon$ is the target error and is aimed at being made small.
>
> As explained in the answers to Reviewers g22n and tau2, we will add discussions that compare our work with the most closely related ones, thus hoping to answer to their questions, and to the clarity comment pointed out in this review.

---

### Meta-Review · Area_Chair_gkPm · 2022-08-27

**Recommendation:** Accept
**Confidence:** Less certain

**Metareview:**

This paper addresses an important issue related to sample complexities for a personalized federated learning (PFL) problem. Its focus is on the case where a large number of agents collaborate to train the PFL problem, and each agent can have local data from a slightly different data distribution.
Author(s) provide both the lower and upper bound on the number of samples needed in order to achieve their goals. They also discuss techniques that allow for achieving an optimal bias-variance trade-off.

Note: please try to incorporate the suggestions and discussions into the camera-ready version.

**Award:**

No

---

### Decision · Program_Chairs · 2022-09-14

Accept